# The Best of Both Worlds: A Framework for Combining Degradation Prediction with High Performance Super-Resolution Networks

**DOI:** 10.3390/s23010419

**Published:** 2022-12-30

**Authors:** Matthew Aquilina, Keith George Ciantar, Christian Galea, Kenneth P. Camilleri, Reuben A. Farrugia, John Abela

**Affiliations:** 1Department of Communications & Computer Engineering, Faculty of ICT, University of Malta, MSD2080 Msida, Malta; 2Deanery of Molecular, Genetic & Population Health Sciences, University of Edinburgh, Edinburgh EH9 3DW, UK; 3Ascent, 90/3, Alpha Centre, Tarxien Road, LQA1815 Luqa, Malta; 4Department of Systems & Control Engineering, Faculty of Engineering, University of Malta, MSD2080 Msida, Malta; 5Department of Computer Information Systems, Faculty of ICT, University of Malta, MSD2080 Msida, Malta

**Keywords:** blind super-resolution, meta-attention, degradation prediction, metadata fusion, iterative prediction, contrastive learning, deep learning

## Abstract

To date, the best-performing blind super-resolution (SR) techniques follow one of two paradigms: (A) train standard SR networks on synthetic low-resolution–high-resolution (LR–HR) pairs or (B) predict the degradations of an LR image and then use these to inform a customised SR network. Despite significant progress, subscribers to the former miss out on useful degradation information and followers of the latter rely on weaker SR networks, which are significantly outperformed by the latest architectural advancements. In this work, we present a framework for combining any blind SR prediction mechanism with any deep SR network. We show that a single lightweight metadata insertion block together with a degradation prediction mechanism can allow non-blind SR architectures to rival or outperform state-of-the-art dedicated blind SR networks. We implement various contrastive and iterative degradation prediction schemes and show they are readily compatible with high-performance SR networks such as RCAN and HAN within our framework. Furthermore, we demonstrate our framework’s robustness by successfully performing blind SR on images degraded with blurring, noise and compression. This represents the first explicit combined blind prediction and SR of images degraded with such a complex pipeline, acting as a baseline for further advancements.

## 1. Introduction

Super-Resolution (SR) is the process by which a Low-Resolution (LR) image is upscaled, with the aim of enhancing both the image’s quality and level of detail. This operation enables the exposure of previously-hidden information which can then subsequently be used to improve the performance of any tasks depending on the super-resolved image. SR is thus highly desirable in a vast number of important applications such as medical imaging [1,2], remote sensing [3,4,5], and in the identification of criminals depicted in Closed-Circuit Television (CCTV) cameras during forensic investigations [6,7].

Single Image SR (SISR) is typically formulated as the restoration of High-Resolution (HR) images that have been bicubically downsampled or blurred and downsampled. On these types of LR images, state-of-the-art (SOTA) SR models can achieve extremely high performance, either by optimising for high pixel fidelity to the HR image [8,9,10,11,12,13], or by improving perceptual quality [14,15,16]. However, real-world images are often affected by additional factors such as sensor noise, complex blurring, and compression [17,18,19], which further deteriorate the image content and make the restoration process significantly more difficult. Moreover, many SR methods are trained on synthetically generated pairwise LR–HR images which only model a subset of the potential degradations encountered in real-world imaging systems [18,20]. As a result, the domain gap between synthetic and realistic data often causes such SR methods to perform poorly in the real world, hindering their practical use [6,18,21].

The field of blind SR is actively attempting to design techniques for image restoration which can deal with more realistic images containing unknown and complex degradations [18]. These methods often break down the problem by first estimating the degradations within an image, after which this prediction is used to improve the performance of an associated SR model. Prediction systems can range from the explicit, such as estimating the shape/size of a specific blur kernel, to the implicit, such as the abstract representation of a degradation within a Deep Neural Network (DNN) [18]. In the explicit domain, significant progress has been made in improving the accuracy and reliability of the degradation parameter estimation process. Recent mechanisms based on iterative improvement [22,23] and contrastive learning [24,25] have been capable of predicting the shape, size and noise of applied blur kernels with little to no error. However, such methods then go on to apply their prediction mechanisms with SR architectures that are smaller and less sophisticated than those used for SOTA non-blind SR.

In this work, we investigate how blind degradation prediction systems can be combined with any SR network with a Convolutional Neural Network (CNN) component, regardless of the network architecture. A robust system for the integration of the blind and SR components would allow for new techniques (both SR architectures or prediction mechanisms) to be immediately integrated and assessed under a blind setting. This would expedite and standardise the blind SR evaluation process, as well as allow any new mechanism to benefit from the latest SOTA architectures without requiring a complete redesign.

In our approach, we use a *metadata insertion block* to link the prediction and SR mechanisms, an operation which interfaces degradation vectors with SR network feature maps. We implement a variety of SR architectures and integrate these with the latest techniques for contrastive learning and iterative degradation prediction. Our results show that by using just a single Meta-Attention (MA) layer [26], high-performance SR models such as the Residual Channel Attention Network (RCAN) [8] and the Holistic Attention Network (HAN) [10] can be infused with degradation information to yield SR results which outperform those of the original blind SR networks trained under the same conditions.

We further extend our premise by performing blind degradation prediction and SR on images with blurring, noise and compression, constituting a significantly more complex degradation pipeline than that studied to-date by other prediction networks [22,23,24,25,27]. We show that, even on such a difficult dataset, our framework is still capable of generating improved SR performance when combined with a suitable degradation prediction system.

The main contributions of this paper are thus as follows:A framework for the integration of degradation prediction systems into SOTA non-blind SR networks.A comprehensive evaluation of different methods for the insertion of blur kernel metadata into CNN SR networks. Specifically, our results show that simple metadata insertion blocks ( such as MA) can match the performance of more complex metadata insertion systems when used in conjunction with large SR networks.Blind SR results using a combination of non-blind SR networks and SOTA degradation prediction systems. These hybrid models show improved performance over both the original SR network and the original blind prediction system.A thorough comparison of unsupervised, semi-supervised and supervised degradation prediction methods for both simple and complex degradation pipelines.The successful application of combined (i) blind degradation prediction and (ii) SR of images, degraded with a complex pipeline involving multiple types of noise, blurring and compression that is more reflective of real-world applications than considered by most SISR approaches.

The rest of this paper is organised as follows: Section 2 provides an overview of related work on general and blind SR, including the methods selected for our framework. Section 3 follows up with a detailed description of our proposed methodology for combining degradation prediction methods with SOTA SISR architectures. Our framework implementation details, evaluation protocol, degradation prediction and SR results are presented and discussed in Section 4. Finally, Section 5 provides concluding remarks and potential areas for further exploration.

## 2. Related Work

A vast number of methods have been proposed for SR, from the seminal Super-Resolution Convolutional Neural Network (SRCNN) [28] to more advanced networks such as RCAN [8], HAN [10], Second-Order Attention Network (SAN) [9], Super-Resolution GAN (SRGAN) [14], and Enhanced SRGAN (ESRGAN) [15] among others. Domain-specific methods have also been implemented, such as those geared for the super-resolution of face images (including Super-Face Alignment Network (Super-FAN) [29] and the methods proposed in [7,30,31,32,33,34,35,36]), and satellite imagery [3,4,37] among others.

Most approaches derive an LR image from a HR image using a *degradation model*, which formulates how this process is performed and the relationship between the LR and HR images. Hence, an overview of common degradation models, including the one used as the basis for the proposed SR framework, is first provided and discussed. Given that the proposed framework combines techniques designed for both *general-purpose SR* and *blind SR*, an overview of popular and SOTA networks for both methodologies will then be provided.

### 2.1. Degradation Models

Numerous works in the literature have focused on the degradation models considered in the SR process, which define how HR images are degraded to yield LR images. However, the formation of an LR image $I^{LR}$ can be generally expressed by the application of a function *f* on the HR image $I^{HR}$, as follows:(1)$$I^{LR}=f\left(I^{HR},\theta_{D}\right)$$
where $\theta_{D}$ is the set of degradation parameters which, in practice, are unknown.

The function *f* can be expanded to consider the general set of degradations applied to $I^{HR}$, yielding the ‘classical’ degradation model as follows [6,18,19,20,22,23,38,39,40,41,42,43]:(2)$$I^{LR}=\left(I^{HR}⊗k\right){↓}_{s}+n$$
where ⊗ represents the convolution operation, *k* is a kernel (typically a Gaussian blurring kernel, although it can also represent other functions such as the Point Spread Function (PSF)), *n* represents additive noise, and ${↓}_{s}$ is a downscaling operation (typically assumed to be bicubic downsampling [18]) with scale factor *s*. However, this model has been criticised for being too simplistic and unable to generalise well to more complex degradations that are found in real-world images, thereby causing substantial performance losses when SR methods based on this degradation model are applied to non-synthetic images [17,19,39]. More complex and realistic degradation types have thus been considered, such as compression (which is typically signal-dependent and non-uniform, in contrast to the other degradations considered [19]), to yield a more general degradation model [17,18,44]: (3)$$I^{LR}={\left(\left(I^{HR}⊗k\right){↓}_{s}+n\right)}_{C}$$
where *C* is a compression scheme such as JPEG.

The aim of SR is then to approximate the inverse function of *f*, denoted by $f^{-1}$. This function can be applied on the LR image ($I^{LR}$) to reverse the degradation process and yield an image ${\hat{I}}^{HR}$ approximating the original image ($I^{HR}$):(4)$${\hat{I}}^{HR}=f^{-1}\left(I^{LR},\theta_{R}\right)\approx I^{HR}$$
where $\theta_{R}$ represents the parameter set defining the reconstruction process. This degradation model forms the basis of the proposed SR framework.

Other works have extended the general model in Equation (Equation 3) to more complex cases. One such work synthesises training pairs using a ‘high-order’ degradation process where the degradation model is applied more than once [16]. The authors of [19] proposed a ‘practical’ degradation model to train the ESRGAN-based BSRNet and BSRGAN models [19], which consider multiple Gaussian blur kernels, downscaling operators, noise levels modelled by Additive White Gaussian Noise (AWGN), processed camera sensor noise types, and quality factors of JPEG compression. Random shuffling of the order in which the degradations are applied is also performed.

Counter-arguments to these complex models have also been made. For instance, the authors of [17] argued that ‘practical’ degradation models as proposed in [16,19] (so called because a wide variety of degradations are considered, reflecting practical real-world applications) may achieve promising results on complex degradations but then ignore easier edge cases. Given that such cases primarily entail combinations of degradation subsets, a gated degradation model is proposed whereby the base degradations to be applied are randomly selected. Since the magnitudes of some degradations in the proposed framework may be reduced to the point where they are practically negligible, the degradation model forming the basis of this work (which is based on the processes defined in Equations (Equation 3) and (Equation 4) as previously discussed) can be said to approximate this gated mechanism. This also means that the proposed framework considers degradations found in practical real-world applications.

### 2.2. Non-Blind SR Methods

Most methods proposed for SR have tended to focus on the case where degradations are assumed to be known, either by designing models for specific degradations or by designing approaches that are able to use Appendix A about the degradations afflicting the image. However, this information is neither estimated nor derived from the corrupted image, limiting the use of such methods in the real world where degradations are highly variable in terms of both their type and magnitude. Despite these limitations, non-blind SR methods have served an important role in enabling more rapid development of new techniques on what is arguably a simpler case of SR. An overview of notable methods will now be given.

The seminal non-blind SR method is considered to be the SRCNN network [28], which was one of the first to use deep learning and CNNs for the task of SR. However, it only consists of three layers and requires the LR image to first be upsampled using bicubic interpolation, and is now outperformed by most modern approaches.

To facilitate the training of a large number of CNN layers, the Residual Network (ResNet) architecture proposed in [45] introduced *skip connections* to directly feed feature maps at any level of the network to deeper layers, a process corresponding to the identity function which deep networks find hard to learn. This counteracted the problem of *vanishing gradients* apparent in classical deep CNN networks, allowing the authors of [45] to expand their network size without impacting training performance. ResNet was extended to SR in [14], to create the Super-Resolution Residual Network (SRResNet) approach that was also used as the basis of a Generative Adversarial Network (GAN)-based approach termed SRGAN.

SRGAN was extended in [15] to yield ESRGAN, which included the introduction of a modified adversarial loss to determine the relative ‘realness’ of an image, and not just a decision as to whether the generated image is ‘real’ or ‘fake’. ESRGAN also introduced a modified VGG-based perceptual loss, which uses feature maps extracted from the VGG residual blocks right before the activation layers to reduce sparsity and better supervise brightness consistency and texture recovery. ESRGAN was further extended in [16] to yield Real ESRGAN (Real-ESRGAN), where the focus was on the implementation of a ‘high-order’ degradation process that allowed the application of the degradation model more than once (in contrast to most works which only apply the model one time).

Enhanced Deep Super-Resolution (EDSR) [46] was also based on ResNet and incorporated observations noted in previous works such as SRResNet, along with other novel contributions that had a large impact on subsequent CNN-based SR models. These included the removal of batch normalisation layers to disable restriction of feature values and reduce the memory usage during training that, in turn, allowed for a greater number of layers and filters to be used.

The RCAN approach proposed in [8] is composed of ‘residual groups’ that each contain a number of ‘channel attention blocks’, along with ‘long’ and ‘short’ skip connections to enable the training of very deep CNNs. The channel attention blocks enable the assignment of different levels of importance of low-frequency information across feature map channels. The concept of attention introduced by RCAN was developed further by other methods such as SAN [9] and HAN [10], which proposed techniques such as channel-wise feature re-scaling and modelling of any inter-dependencies among channels and layers.

Recently, vision transformers applied to SR were also proposed, such as the Encoder-Decoder-based Transformer (EDT) [47], Efficient SR Transformer (ESRT) [48], and the Swin Image Restoration (SwinIR) [12] approach that is based on the Swin Transformer [49]. Approaches such as Efficient Long-Range Attention Network (ELAN) [13] and Hybrid Attention Transformer (HAT) [50], which attempt to combine CNN and transformer architectures, have also been proposed with further improvements in SR performance.

An extensive exposition of generic non-blind SR methods can be found in [51,52].

### 2.3. Blind SR Methods

Although numerous SR methods have been proposed, a substantial number of approaches tend to employ the classical degradation model (in Equation (Equation 2)). Besides not being quite reflective of real-world degradations (as discussed in Section 2.1), a substantial number of approaches also assume that the degradations afflicting an image are known. This is largely not the case, so that such approaches tend to exhibit noticeable performance degradation on images found ‘in-the-wild’.

As a result, *Blind SR* methods have been designed for better robustness when faced with such difficult and unknown degradations, making them more suitable for real-world applications. There exist several types of blind SR methods, based on the type of data used and how they are modelled [18]. An overview of the various types of approaches and representative methods will now be provided.

#### 2.3.1. Approaches Utilising Supplementary Attributes for SR

Early work focused on the development of methods that are directly supplied with ground-truth information on degradations. The focus then turned on developing techniques to best utilise this degradation information (as opposed to non-blind SR methods which do not use any form of Appendix A). Approaches of this kind generally consider the classical degradation model [18].

Notable methods incorporating metadata information in networks include Super-Resolution network for Multiple Degradations (SRMD) [39], Unified Dynamic Convolutional Network for Variational Degradations (UDVD) [53], the Deep Plug-and-Play SR (DPSR) framework, and the approach in [54]. Each of these approaches showed that SR networks could improve their performance by using this degradation information. Frameworks enabling the extension of existing non-blind SR methods to use degradation information have also been proposed, such as the ‘meta-attention’ approach in [26] and Conditional hyper-network framework for SR with Multiple Degradations (CMDSR) proposed in [55].

These methods clearly show the plausibility of improving SR performance with degradation metadata. However, apart from requiring some means of obtaining the degradation information, such methods are also highly reliant on the quality of the information input to the networks, which is not a trivial task. Moreover, any deviations in the estimated inputs can lead to kernel mismatches that may be detrimental to the SR performance [18,23,38].

#### 2.3.2. Iterative Kernel Estimation Methods

Blur kernel estimation during the SR process is one of the most common blind SR prediction tasks, and alleviates the problem of kernel mismatches present in methods such as SRMD as described above. Often, iterative mechanisms are applied for direct kernel estimation. One such method is Iterative Kernel Correction (IKC) [22], which leverages the observation that kernel mismatch tends to produce regular patterns by estimating the degradation kernel and correcting it in an iterative fashion using a corrector network. In this way, an acceptable result is progressively approached. The authors of [22] also proposed a non-blind SR network, named Spatial Feature Transform Multiple Degradations (SFTMD), which was shown to outperform existing methods (such as SRMD) that insert blur kernel metadata into the SR process.

The Deep Alternating Network (DAN) [23] method (also known as DANv1) and its updated version DANv2 [38] build upon the IKC approach, by combining the SR and kernel corrector networks within a single end-to-end trainable network. The corrector was also modified to use the LR input conditioned on intermediate super-resolved images, instead of conditioning these images on the estimated kernel as done in IKC. The Kernel-Oriented Adaptive Local Adjustment network (KOALAnet) [56] is able to adapt to spatially-variant characteristics within an image, which allows a distinction to be made between blur caused by undesirable effects, and between blur introduced intentionally for aesthetic purposes (e.g., the Bokeh effect). However, given that such methods still rely on kernel estimation, they also exhibit poor performance when evaluated on images having different degradations than those used to train the model.

#### 2.3.3. Training SR Models on a Single Image

Another group of methods such as KernelGAN [57] and Zero-Shot SR (ZSSR) [58] use intra- and inter-scale recurrence of patches, based on the internal statistics of natural images, to construct an individual model for each input LR image. Hence, the data used for training is that which is present internally within the image being super-resolved, circumventing the need to use an external dataset of images.

Such methods tend to assume that a downscaled version of a patch within a LR image should have a similar distribution to the patch in the original LR image. However, the assumption of recurring patches within and across scales may not hold true for all images (such as those containing a wide variety of content) [18].

#### 2.3.4. Implicit Degradation Modelling

Modelling an explicit combination of multiple degradation types can be a very complex task on images found ‘in-the-wild’. Hence, some approaches have also attempted to implicitly model the degradation process by comparing the data distribution of real-world LR image sets with synthetically generated ‘clean’ datasets (containing limited or no degradations) [18]. Methods are typically based on GANs, such as Cycle-in-Cycle GAN (CinCGAN) [59] and the approaches in [60,61], and do not require a HR reference for training.

However, one of the drawbacks of this type of method is that they tend to require vast amounts of data, which may not always be available. Some approaches, such as Degradation GAN [62] and Frequency Separation for real-world SR (FSSR) [63], attempt to counteract this issue by learning the HR-to-LR degradation process to generate realistic LR samples that can be used during the SR model training. However, most models designed for implicit degradation modelling use GANs which are known to be hard to train and may introduce fake textures or artefacts that can be detrimental for some real-world applications [18].

#### 2.3.5. Contrastive Learning

In the image classification domain, DNNs are known to be highly capable of learning invariant representations, enabling the construction of good classifiers [64]. However, it has been argued that DNNs are actually too eager to learn invariances [64]. This is because they often learn only the features necessary to discriminate between classes but then fail to generalise well to new unseen classes when employing a supervised setting in what is known as *supervision collapse* [64,65]. Indeed, the ubiquitous cross-entropy loss used to train supervised deep classifier models has received criticism for several shortcomings [66], such as its sensitivity to noisy labels [67,68] and the possibility of poor margins [69,70,71].

*Contrastive learning* techniques, mostly developed in the Natural Language Processing (NLP) domain, have recently seen a resurgence and have driven significant advances in self-supervised representation learning in an attempt to mitigate these issues [66]. Contrastive learning is a self-supervised approach where models are trained by comparing and contrasting ‘positive’ image pairs with ‘negative’ pairs [72]. Positive images can be easily created by applying augmentations to a source image (e.g., flipping, rotations, colour jitter etc.). In the SR domain, positive samples are typically patches extracted from within the same image, while crops taken from other images are labelled as negative examples [24,72,73].

One such contrastive learning-based approach is Momentum Contrast (MoCo), proposed in [73] for the tasks of object detection, classification, and segmentation. MoCo employs a large queue of data samples to enable the use of a dictionary (containing samples observed in preceding mini-batches) which is much larger than the mini-batch size. However, since a large dictionary also makes it intractable to update the network parameters using back-propagation, a momentum update which tightly controls the parameters’ rate of change is also proposed. MoCo was extended in [74] to yield MoCov2, based on design improvements proposed for SimCLR [72]. The two main modifications constitute the replacement of the fully-connected layer at the head of the network with an Multi-Layer Perceptron (MLP) head, and the inclusion of blur augmentation.

Supervised MoCo (SupMoCo) [64] was also proposed as an extension of MoCo, whereby class labels are additionally utilised to enable intra-class variations to be learnt whilst retaining knowledge on distinctive features acquired by the self-supervised components. SupMoCo was shown to outperform the Supervised Contrastive (SupCon) approach [66], which also applied supervision to SimCLR [72].

Such self-supervised methods have seen limited use in the SR domain thus far. However, the promising performance demonstrated in other domains could encourage further research and development in the SR arena. One of the first SR networks to use contrastive learning for blind SR was the Degradation-Aware SR (DASR) network [24], where an unsupervised content-invariant degradation representation is obtained in a latent feature space where the mutual information among all samples is maximised.

In [25], Implicit Degradation Modelling Blind SR (IDMBSR), considers the degrees of difference in degradation between a query image and negative exemplars, in order to determine the amount of ‘push’ to exert. Specifically, the greater the difference between a query and a negative example, the greater the push. In this way, degradation information is used as weak supervision of the representation learning to improve the network’s ability to characterise image degradations.

#### 2.3.6. Other Methodologies

The mechanisms discussed in this section are some of the most prominent in the blind SR literature. However, many other modalities exist which do not neatly fall into any single category. One such method is Mixture of Experts SR (MoESR) [42], where a panel of expert predictors is used to help optimise the prediction system for blur kernels of different shapes and sizes. A comprehensive overview of the state of blind SR research can be found in [18].

### 2.4. Conclusions

As described above, there exist numerous algorithms, architectures, and frameworks designed to perform the task of SR, both for non-blind and blind scenarios. However, current approaches for both tasks suffer from major drawbacks. On the one hand, non-blind SR architectures have been optimised for high performance on synthetic data, but under-perform in practical degradation scenarios. On the other hand, blind SR networks and degradation prediction systems are more capable of dealing with real-world images but use weaker and more limited architectures that restrict their performance ceiling. These observations serve as the motivation to *combine the best of both worlds*, chiefly to leverage the performance of non-blind SR networks with the practical applications of degradation prediction systems.

Given the prominence of iterative and contrastive methods in blind SR, a number of these mechanisms were selected and implemented within the proposed framework. The principles of SupMoCo were also applied to construct a more controllable self-supervised contrastive learning function (Section 3.5.2). However, in principle, any degradation estimation system could be coupled to any SR model, using the framework described in the rest of this paper.

## 3. Methodology

### 3.1. Degradation Model

For all our analyses, we follow the degradation model described in Equation (Equation 3) that encompasses some of the most common degradations found in the real world, namely blurring, downsampling, instrument noise, and compression. The task of any SR algorithm is to then essentially reverse the degradation process (as shown in Equation (Equation 4)) afflicting an LR image $I^{LR}$ to yield an image ${\hat{I}}^{HR}$ that approximates the original HR image ($I^{HR}$).

For the LR–HR pairs used for training and testing, a variety of different operations across each type of degradation are applied, with the full details provided in Section 4.1. Given that the order, type, and the parameters of the degradations are known in advance, the task of our degradation prediction models is significantly easier than the fully blind case with completely unknown degradations. However, the degradation prediction principles of each model could easily be extended to more complex and realistic degradation pipelines [16,19] through adjustments to the degradation modelling process. That said, a variety of degradation magnitudes emulating the ‘gated’ degradation pipeline as proposed in [17] is considered, which is more reflective of real-world scenarios as discussed in Section 2.1. Indeed, it is shown in Section 4.7 that our models are still capable of dealing with test images degraded in the real world despite the easier conditions in the training datasets.

### 3.2. Framework for Combining SR Models with a Blind Predictor

Our proposed general framework for combining blind SR prediction mechanisms with non-blind SR models aims to amplify the strengths of both techniques with minimal architectural changes on either side. In most cases, explicit blind prediction systems generate vectors to describe the degradations present in an image. On the other hand, the vast majority of SR networks feature convolutional layers and work in image space rather than in a vector space. To combine the two mechanisms, the prediction model was kept separate from the SR core and the two were bridged using a *metadata insertion block*, as shown in Figure 1. This makes it relatively simple for different prediction or SR blocks to be swapped in and out, while keeping the overall framework unchanged. We considered a variety of options for each of the three main components of the framework. The metadata insertion and degradation prediction mechanisms selected and adjusted for our framework are discussed in the remaining sections of the methodology, while the chosen SR core networks are provided in Section 4.1.

### 3.3. Metadata Insertion Block

The metadata insertion block plays an essential role in our framework, since it converts a degradation vector into a format compatible with CNN-based SR networks, and ensures that this information is fully utilised throughout the SR process. Despite its importance, the inner workings of the mechanism are poorly understood since CNN-based models (SR or otherwise) are notoriously difficult to interpret. In fact, multiple methods for combining vectors and images within CNNs have been proposed and these vary significantly in complexity and size without a clear winner being evident. We selected and tested some of the most effective mechanisms in the literature within our framework, of which the following is a brief description of each (with a graphical depiction provided in Figure 2):**SRMD-style [39] (Figure 2A):** An early adopter of metadata insertion in SR, the SRMD technique involves the transformation of vectors as additional pseudo-image channels. Each element of the input degradation vector is expanded (by repeated tiling) into a 2D array, with the same dimensions as the input LR image. These pseudo-channels are then fed into the network along with the real image data, ensuring that all convolutional filters in the first layer have access to the degradation information. Other variants of this method, which include directly combining the pseudo-channels with CNN feature maps, have also been proposed [23]. The original SRMD network used this method for Principal Component Analysis (PCA)-reduced blur kernels and noise values. In our work, we extended this methodology to all degradation vectors considered.**MA [26] (Figure 2B):** MA is a trainable channel attention block which was proposed as a way to upgrade any CNN-based SR network with metadata information. Its functionality is simple—an input vector is stretched to the same size as the number of feature maps within a target CNN network using two fully-connected layers. Each vector element is normalised to lie in the closed interval [0,1] and then applied to selectively amplify its corresponding CNN feature map. MA was previously applied for PCA-reduced blur kernels and compression quality factors only. We extended this mechanism to all degradation parameters considered, by combining them into an input vector which is then fed into the MA block. The fully-connected layer sizes were then expanded as necessary to accommodate the input vector.**SFT [22] (Figure 2C):** The SFT block is based on the SRMD concept but with additional layers of complexity. The input vector is also stretched into pseudo-image channels, but these are added to the feature maps within the network rather than the actual original image channels. This combination of feature maps and pseudo-channels are then fed into two separate convolutional pathways, one of which is multiplied with the original feature maps and the other added on at the end of the block. This mechanism is the largest (in terms of parameter count due to the number of convolutional layers) of those considered in this paper. As with SRMD, this method has only been applied for blur kernel and noise parameter values, and we again extended the basic concept to incorporate all degradation vectors considered.**Degradation-aware (DA) block [24] (Figure 2D):** The DA block was proposed in combination with a contrastive-based blind SR mechanism for predicting blur kernel degradations. It uses two parallel pathways, one of which amplifies feature maps in a manner similar to MA, while the other explicitly transforms vector metadata into a 3D kernel, which is applied on the network feature maps. This architecture is highly specialised to kernel-like degradation vectors, but could still be applicable for general degradation parameters given its dual pathways. We extended the DA block to all degradation vectors as we did with the MA system.**Degradation-Guided Feature Modulation Block (DGFMB) [25] (Figure 2E):** This block was conceived as part of another contrastive-based network, again intended for blur and noise degradations. The main difference here is that the network feature maps are first reduced into vectors and concatenated with the degradation metadata in this form, rather than in image space. Once the vectors are concatenated, a similar mechanism to MA is used to selectively amplify the output network feature maps. As before, we extended this mechanism to other degradation parameters by combining these into an input vector.

Many of the discussed metadata insertion mechanisms were initially introduced as repeated blocks which should be distributed across the entirety of an SR network. However, this can significantly increase the complexity (in both size and speed) of a network as well as make implementation difficult, given the variety of network architectures available. Our testing has shown that, in most cases, simply adding one metadata-insertion block at the front of the network is enough to fully exploit the degradation vector information (results in Section 4.2). Further implementation details of each block are provided in Section 4.1.

### 3.4. Degradation Prediction—Iterative Mechanism

The simplest degradation prediction system tested for our framework is the DAN iterative mechanism proposed in [23]. This network consists of two convolutional sub-modules—a *restorer*, in charge of the super-resolution step and an *estimator*, which predicts the blur kernel applied on an LR image (in PCA reduced form). Both modules are co-dependent; the restorer produces an SR image based on the degradation prediction while the estimator makes a degradation prediction informed by the SR image. By repeatedly alternating between the two modules, the results of both can be iteratively improved. Furthermore, both networks can be optimised simultaneously by back-propagating the error of the SR and degradation estimates.

The iterative mechanism is straightforward to introduce into our framework. We implemented the estimator module from DAN directly, and then coupled its output with a core SR network through a metadata insertion block (Figure 1). While the authors of DAN only considered blur kernels in their work, it should be possible to direct the estimator to predict the parameters of any specified degradation directly. We tested this hypothesis for both simple and complex degradations, the results of which are provided in Section 4.3 and Section 4.5.

### 3.5. Degradation Prediction—Contrastive Learning

Contrastive learning is another prominent method for degradation estimation in blind SR. We considered three methods for contrastive loss calculation: one completely unsupervised and two semi-supervised techniques. These mechanisms have been illustrated in Figure 3A, and are described in further detail hereunder.

#### 3.5.1. MoCo—Unsupervised Mechanism

Contrastive learning for blind SR was first proposed in [24], where the authors used the MoCo [73] mechanism to train convolutional encoders that are able to estimate the shape and noise content in blur kernels applied on LR images. The encoder is taught to generate closely-matched vectors for images with identical or similar degradations (e.g., equally sized blur kernels) and likewise generate disparate vectors for vastly different degradations. These encoded vectors, while not directly interpretable, can be utilised by a downstream SR model to inform the SR process. The proposed MoCo encoder training mechanism works as follows:Two identical encoders are instantiated. One acts as the ‘query’ encoder and the other as the ‘key’ encoder. The query encoder is updated directly via backpropagation from computed loss, while the key encoder is updated through a momentum mechanism from the query encoder.The encoders are each directed to generate a degradation vector from one separate square patch per LR image, as shown on the right-hand side of Figure 3A. The query encoder vector is considered as the reference for loss calculation, while the key encoder vector generated from the second patch acts as a positive sample. The training objective is to drive the query vector to become more similar to the positive sample vector, while simultaneously repelling the query vector away from encodings generated from all other LR images (negative samples).Negative samples are generated by placing previous key encoder vectors from distinct LR images into a queue. With both positive and negative encoded vectors available, an InfoNCE-based [75] loss function can be applied:
(5)$$L_{MOCO}=\frac{1}{B}\left[\sum_{i=1}^{B}-\log\frac{\exp\left(f_{q}\left(x_{i}^{1}\right)\cdot f_{k}\left(x_{i}^{2}\right)/\tau \right)}{\exp\left(f_{q}\left(x_{i}^{1}\right)\cdot f_{k}\left(x_{i}^{2}\right)/\tau \right)+\sum_{j=1}^{N_{\mathrm{queue}\;}}\exp\left(f_{q}\left(x_{i}^{1}\right)\cdot q^{j}/\tau \right)}\right]$$
where $f_{q}$ and $f_{k}$ are the query and key encoders, respectively, $x_{i}^{1}$ is the first patch from the *i*th image in a batch (batch size *B*), $q^{j}$ is the *j*th entry of the queue of size $N_{queue}$ and τ is a constant temperature scalar. With this loss function, the query encoder is updated to simultaneously move its prediction closer to the positive encoding, and farther away from the negative set of encodings. This push-pull effect is depicted by the coloured dotted lines within the MoCo box in Figure 3A. This loss should enable the encoder to distinguish between the different degradations present in the positive and negative samples.

In [24], only one positive patch is used per input image. However, this can be easily extended (generalised) to multiple positive patches through the following modifications to the loss function (shown in blue): (6)$$L_{MOCO}=\frac{1}{B\times P^{i}}\left[\sum_{i=1}^{B}-\log\frac{\sum_{l=1}^{P^{i}}\exp\left(f_{q}\left(x_{i}^{1}\right)\cdot f_{k}\left(x_{i}^{l}\right)/\tau \right)}{\sum_{l=1}^{P^{i}}\exp\left(f_{q}\left(x_{i}^{1}\right)\cdot f_{k}\left(x_{i}^{l}\right)/\tau \right)+\sum_{j=1}^{N_{\mathrm{queue}\;}}\exp\left(f_{q}\left(x_{i}^{1}\right)\cdot q^{k}/\tau \right)}\right]$$
where $P^{i}$ is the number of positive patches for the $i^{\mathrm{th}}$ image in a batch. For most of our tests, we retain just one positive patch (using the loss from Equation (Equation 5)) to match with [24], unless indicated.

#### 3.5.2. SupMoCo—Semi-Supervised Mechanism

In the image classification domain, advances into semi-supervised contrastive learning have resulted in further performance improvements over MoCo. One such mechanism, SupMoCo [64], provides more control over the contrastive training process. In essence, all encoded vectors can be assigned a user-defined label, including the query vector. With these labels, the contrastive loss function can be directed to push the query vector towards all the key vectors sharing the same label, while repelling it away from those that have different labels, as shown by the SupMoCo column of Figure 3A. To do this, the authors of [64] added a label-based loss component to Equation (Equation 6) which also considers the positive samples in the queue (highlighted in blue): (7)$$L_{SUPMOCO}=\frac{1}{B\times F^{i}}\left[\sum_{i=1}^{B}-\log\frac{\sum_{l=1}^{P^{i}}\exp\left(f_{q}\left(x_{i}^{1}\right)\cdot f_{k}\left(x_{i}^{l}\right)/\tau \right)+\sum_{m=1}^{Q^{i}}\exp\left(f_{q}\left(x_{i}^{1}\right)\cdot f_{k}\left(x_{i}^{m}\right)/\tau \right)}{\sum_{l=1}^{P^{i}}\exp\left(f_{q}\left(x_{i}^{1}\right)\cdot f_{k}\left(x_{i}^{l}\right)/\tau \right)+\sum_{j=1}^{N_{\mathrm{queue}\;}}\exp\left(f_{q}\left(x_{i}^{1}\right)\cdot q^{k}/\tau \right)}\right]$$
where $Q^{i}$ is the number of samples in the queue with the same label as the query vector and $F^{i}=P^{i}+Q^{i}$.

This system allows more control of the trajectory of the contrastive loss, reducing inaccuracies while pushing the encoder to recognise patterns based on the labels provided. For our degradation pipeline, we implemented a decision tree which assigns a unique label to each possible combination of degradations. In brief, a label is assigned to each degradation based on a number of factors:Blur kernel typeBlur kernel size; either low/high, which we refer to as *double* precision (2 clusters per parameter), or low/medium/high, which we refer to as *triple* precision (3 clusters per parameter), classification.Noise typeNoise magnitude (either a double or triple precision classification)Compression typeCompression magnitude (either a double or triple precision classification)

An example of how this decision tree would work for labelling of compression type and magnitude is provided in Figure 4.

The decision tree labelling should push the encoder to identify the presence of different degradation classes more quickly than in the unsupervised case. Aside from the labelling system, SupMoCo is trained in an identical fashion to MoCo, including the usage of momentum to update the key encoder. A full description of all degradations applied in our pipeline is provided in Section 4.1.

#### 3.5.3. WeakCon—Semi-Supervised Mechanism

Another semi-supervised contrastive paradigm (which we refer to as WeakCon) has been proposed in [25]. Instead of assigning discrete labels to each degradation, the authors propose a system for modulating the strength of the contrastive loss. By calculating the difference in degradation magnitudes between query and negative samples, the negative contrastive push can be increased or decreased, according to how different the degradations are. This weighted push is illustrated in the WeakCon column of Figure 3A. In [25], the authors utilise , the Euclidean distance between query and selected negative sample blur kernel widths and noise sigmas to calculate a weight *w*. Using this weighting (highlighted in blue), the authors updated the InfoNCE-style [75] loss function in Equation (Equation 6) so that the contrastive loss can be weakly-supervised as follows:(8)$$L_{WEAKCON}=\frac{1}{B\times P^{i}}\left[\sum_{i=1}^{B}-\log\frac{\sum_{l=1}^{P^{i}}\exp\left(f_{q}\left(x_{i}^{1}\right)\cdot f_{k}\left(x_{i}^{l}\right)/\tau \right)}{\sum_{l=1}^{P^{i}}\exp\left(f_{q}\left(x_{i}^{1}\right)\cdot f_{k}\left(x_{i}^{l}\right)/\tau \right)+\sum_{j=1}^{N_{\mathrm{queue}\;}}w_{ij}\exp\left(f_{q}\left(x_{i}^{1}\right)\cdot q^{k}/\tau \right)}\right]$$
where $w_{ij}$ indicates the distance between the degradations of negative sample *j* and query sample *i*. Since the original work in [25] focused only of isotropic Gaussian kernels and Gaussian noise, we extend this special case to other degradations by similarly calculating the Euclidean distance between degradation vectors containing blur widths in both horizontal and vertical directions, noise scales and compression qualities.

#### 3.5.4. Direct Regression Attachment

While contrastive representations can be visualised using dimensionality reduction, it is difficult to quantify their prediction accuracy with respect to the true degradation parameters. To provide further insight into the training process, we attach a further set of fully-connected layers to the contrastive encoder, as shown in Figure 3B. These layers are set to directly transform the contrastive vector into the magnitudes of the various degradations being estimated. A regression loss (L1 loss between predicted vector and target degradation magnitudes) can also be introduced as an additional supervised element. This direct parameter prediction can be easily quantified into an estimation error, which can help track training progress. We train various models with and without these extra layers, with the details provided in Section 4.

### 3.6. Extensions to Degradation Prediction

In both the iterative and contrastive cases, our prediction mechanisms are centred around general degradation parameter prediction and, as such, could be extended to any degradation which can be parameterised in some form. Alternatively, degradations could be represented in vector form through the use of dimensionality reduction techniques (as is often done with blur kernels, on which PCA is applied). Dimensionality reduction can also be used as an imperfect view of the differences between contrastive vectors encoded for different degradations. We provide our analyses of the contrastive process in Section 4.3 and Section 4.5.

## 4. Experiments & Results

### 4.1. Implementation Details

#### 4.1.1. Datasets and Degradations

For analysis purposes, we created two LR degradation pipelines:**Simple pipeline (blurring and downsampling):** For our metadata insertion screening and blind SR comparison, we used a degradation set of just Gaussian blurring and bicubic downsampling corresponding to the ‘classical’ degradation model described in Equation (Equation 2). Apart from minimising confounding factors, this allowed us to make direct comparisons with pre-trained models provided by the authors of other blind SR networks. For all scenarios, we used only 21 × 21 isotropic Gaussian kernels with a random width (σ) in the range [0.2,3] (as recommended in [16]), and ×4 bicubic downsampling. The σ parameter was then normalised to the closed interval [0,1] before it was passed to the models.**Complex pipeline:** In our extended blind SR training schemes, we used a full degradation pipeline as specified in Equation (Equation 3), i.e., sequential blurring, downsampling, noise addition and compression. For each operation in the pipeline, a configuration was randomly (from a uniform distribution) selected from the following list:
−*Blurring:* As proposed in [16], we sampled blurring from a total of 7 different kernel shapes: iso/anisotropic Gaussian, iso/anisotropic generalised Gaussian, iso/anisotropic plateau, and sinc. Kernel σ values (both vertical and horizontal) were sampled from the range [0.2,3], kernel rotation ranged from −π to π (all possible rotations) and the shape parameter β ranged from [0.5,8] for both generalised Gaussian and plateau kernels. For sinc kernels, we randomly selected the cutoff frequency from the range [π/5,π]. All kernels were set to a size of 21 × 21 and, in each instance, the blur kernel shape was randomly selected, with equal probability, from the 7 available options. For a full exposition on the selection of each type of kernel, please refer to [16].−*Downsampling:* As in the initial model screening, we again retained ×4 bicubic downsampling for all LR images.−*Noise addition:* Again following [16], we injected noise using one of two different mechanisms, namely Gaussian (signal independent read noise) and Poisson (signal dependent shot noise). Additionally, the noise was either independently added to each colour channel (colour noise), or applied to each channel in an identical manner (grey noise). The Gaussian and Poisson mechanisms were randomly applied with equal probability, grey noise was selected with a probability of 0.4, and the Gaussian/Poisson sigma/scale values were randomly sampled from the ranges [1,30] and [0.05,3], respectively.−*Compression:* We increased the complexity of compression used in previous works by randomly selecting from either JPEG or JM H.264 (version 19) [76] compression at runtime. For JPEG, a quality value was randomly selected from the range [30,95] (following [16]). For JM H.264, images were compressed as single-frame YUV files where a random I-slice Quantization Parameter (QPI) was selected from the range [20,40], as discussed in [26].

To allow for a fair (and direct) comparison to other works, the training, validation and testing HR datasets we selected are identical to those used in the SR works we used as baselines or comparison points (Section 4.1.2). Thus, all our models were trained on LR images generated from HR images of DIV2K [77] (800 images) and Flickr2K [78] (2650 images). Validation and best model selection were performed on the provided DIV2K validation set (100 images).

For final results comparison, the standard SR test sets Set5 [79], Set14 [80], BSDS100 [81], Manga109 [82] and Urban100 [83] were utilised. For these test images, the parameters of each degradation were explicitly selected. The exact degradation details for each scenario are specified in all the tables and figures presented. The super-resolved images were compared with the corresponding target HR images using several metrics during testing and validation, namely Peak Signal-to-Noise Ratio (PSNR), Structural SIMilarity index (SSIM) [84] (direct pixel comparison metrics) and Learned Perceptual Image Patch Similarity (LPIPS) [85] (perceptual quality metric). In all cases, images were first converted to YCbCr, and the Y channel used to compute metrics.

The degradation pipelines and further implementation details are available in our linked PyTorch [86] codebase (https://github.com/um-dsrg/RUMpy).

#### 4.1.2. Model Implementation, Training and Validation

Due to the diversity of the models investigated in this work, a number of different training and validation schemes were followed depending on the task and network being investigated:**Non-blind SR model training**: For non-blind model training, we initialised the networks with the hyperparameters as recommended by their authors, unless otherwise specified. All models were trained from scratch on LR–HR pairs generated from the DIV2K and Flickr2K datasets using either the simple or complex pipelines. For the simple pipeline, one LR image was generated from each HR image. For the complex pipeline, five LR images were generated per HR image to improve the diversity of degradations available. In both cases, the LR image set was generated once and used to train all models. All simple pipeline networks were trained for 1000 epochs, whereas the complex pipeline networks were trained for 200 epochs to ensure fair comparisons (since each epoch contains 5 times as many samples as in the simple case). This training duration (in epochs) was chosen as a compromise between obtaining meaningful results and keeping the total training time low.For both pipelines, training was carried out on 64 × 64 LR patches. The Adam [87] optimiser was used. Variations in batch size and learning rate scheduling were made for specific models as necessary in order to ensure training stability and limit Graphical Processing Unit (GPU) memory requirements. The configurations for the non-blind SR models tested are as follows:−RCAN [8] and HAN [10]: We selected RCAN to act as our baseline model as a compromise between SR performance and architectural simplicity. To push performance boundaries further, we also trained and tested HAN as a representative SOTA pixel-quality CNN-based SR network. For these models, the batch size was set to 8 in most cases, and a cosine annealing scheduler [88] was used with a warm restart after every 125,000 iterations and an initial learning rate of 1 × 10^−4^. Training was driven solely by the L1 loss function which compares the SR image with the target HR image. After training, the epoch checkpoint with the highest validation PSNR was selected for final testing.−Real-ESRGAN [16]: We selected Real-ESRGAN as a representative SOTA perceptual quality SR model. The same scheme described for the original implementation was used to train this network. This involved two phases: (i) a pre-training stage where the generator was trained with just an L1 loss, and (ii) a multi-loss stage where a discriminator and VGG perceptual loss network were introduced (further details are provided in [16]). We pre-trained the model for 715 and 150 epochs (which match the pretrain:GAN ratio as originally proposed in [16]) for the simple and complex pipelines, respectively. In both cases, the pre-training optimiser learning rate was fixed at $2\times 10^{-4}$, while the multi-loss stage involved a fixed learning rate of $1\times 10^{-4}$. A batch size of 8 was used in all cases. After training, the model checkpoint with the lowest validation LPIPS score in the last 10% of epochs was selected for testing.−ELAN [13]: We also conducted a number of experiments with ELAN, a SOTA transformer-based model. For this network a batch size of 8 and a constant learning rate of $2\times 10^{-4}$ were used in all cases. As with RCAN and HAN, the L1 loss was used to drive training and the epoch checkpoint with the highest validation PSNR was selected for final testing.**Iterative Blind SR**: Since the DAN iterative scheme requires the SR image to improve its degradation estimate, the predictor model needs to be trained simultaneously with the SR model. We used the same CNN-based predictor network described in DANv1 [23] for our models and fixed the iteration count to four in all cases (matching the implementation as described in [23]). We coupled this predictor with our non-blind SR models using the framework described in Section 3.2. We trained all DAN models by optimising for the SR L1 loss (identical to the non-blind models) and an additional L1 loss component comparing the prediction and ground-truth vectors. Target vectors varied according to the pipeline, the details of each are provided in their respective results sections. For each specific model architecture, the hyperparameters and validation selection criteria were all set to be identical to that of the base, non-blind model. The batch size for all models was adjusted to 4 due to the increased GPU memory requirements needed for the iterative training scheme. Accordingly, whenever a warm restart scheduler was used, the restart point was adjusted to 250,000 iterations (to maintain the same total number of iterations as performed by the other models that utilised a batch size of 8 for 125,000 iterations).Additionally, we also trained the original DANv1 model from scratch, using the same hyperparameters from [23] and the same validation scheme as the other DAN models. The batch size was also fixed to 4 in all cases.**Contrastive Learning**: We used the same encoder from [24] for most of our contrastive learning schemes. This encoder consists of a convolutional core connected to a set of three fully-connected layers. During training, we used the output of the fully-connected layers (*Q*) to calculate loss values (i.e., $f_{q}$ and $f_{k}$ in Section Equation 5) and update the encoder weights, following [24]. Before coupling the encoder with an SR network, we first pre-trained the encoder directly. For this pre-training, the batch size was set to 32 and data was generated online, i.e., each LR image was synthesised on the fly at runtime. All encoders were trained with a constant learning rate of $1\times 10^{-3}$, a patch size of 64 × 64 and the Adam optimiser. The encoders were trained until the loss started to plateau and t-Distributed Stochastic Neighbour Embedding (t-SNE) clustering of degradations generated on a validation set composed of 400 images from CelebA [89] and BSDS200 [81] was clearly visible (more details on this process are provided in Section 4.3.1). In all cases, the temperature hyperparameter, momentum value, queue length, and encoder output vector size were set to 0.07, 0.999, 8192 and 256, respectively, (matching the models from [24]).After pre-training, each encoder was coupled to non-blind SR networks using the framework discussed in Section 3.2. For standard encoders, the *encoding* (i.e., the output from the convolutional core that bypasses the fully-connected layers) is typically fed into metadata insertion blocks directly, unless specified. For encoders with a regression component (see Figure 2B), the dropdown output is fed to the metadata insertion block instead of the encoding. The combined encoder + SR network was then trained using the same dataset and hyperparameters as the non-blind case. The encoder weights were frozen and no gradients were generated for the encoding at runtime, unless specified.

In our analysis, we use the simple pipeline as our primary blind SR task and the complex pipeline as an extension scenario for the best performing methods. In Section 4.2 we discuss our metadata insertion block testing, while Section 4.3 and Section 4.4 present our degradation prediction and SR analysis on the simple pipeline respectively. Section 4.5 and Section 4.6 follow up with our analysis on the complex pipeline and Section 4.7 presents some of our blind SR results on real-world degraded images.

### 4.2. Metadata Insertion Block Testing

To test and compare the various metadata insertion blocks selected, we implemented each block into RCAN, and trained a separate model from scratch on our simple pipeline dataset. Each metadata insertion block was given either the real blur kernel width (normalised in the range [0,1]) or the PCA-reduced kernel representation, for each LR image. The PSNR test results for each model have been compiled in Table 1, and plotted in a comparative bar graph in Figure 5. The SSIM results are also available in the Appendix A.

From the results, it is evident that metadata insertion provides a significant boost to performance across the board. Somewhat surprisingly, the results also show that no single metadata insertion block has a clear advantage over the rest. Every configuration tested, including those where multiple metadata insertion blocks are provided, produces roughly the same level of performance with only minor variations across dataset/degradation combinations. This outcome suggests that each metadata block is producing the same amount of useful information from the input kernel. Further complexity, such as the DA block’s kernel transformation or the SFT/DGFMB feature map concatenation, provides no further gain in performance. Even adding further detail to the metadata, such as by converting the full blur kernel into a PCA-reduced vector, provides no performance gains. This again seems to suggest that the network is capable of extrapolating the kernel width to the full kernel description, without requiring any additional data engineering. Furthermore, adding just a single block at the beginning of the network appears to be enough to inform the whole network, with additional layers providing no improvement (while a decrease in performance is actually observed in the case of DA). We hypothesise that this might be due to the fact that degradations are mostly resolved in the earlier low-frequency stages of the network.

Given that all metadata insertion blocks provide almost identical performance, we selected a single MA block for our blind SR testing, given its low overhead and simplicity with respect to the other approaches. While it is clear the more complex metadata insertion blocks do not provide increased performance on this dataset, it is still possible that they might provide further benefit if other types of metadata are available.

### 4.3. Blur Kernel Degradation Prediction

To test our degradation prediction mechanisms, we evaluated the performance of these methods on a range of conditions and datasets.

#### 4.3.1. Contrastive Learning

For contrastive learning methods, the prediction vectors generated are not directly interpretable. This makes it difficult to quantify the accuracy of the prediction without some form of clustering/regression analysis. However, through the use of dimensionality reduction techniques such as t-SNE [90], the vectors can be easily reduced to 2-D, which provides an opportunity for qualitative screening of each model in the form of a graph.

We trained separate encoders using our three contrastive algorithms on the simple degradation pipeline. The testing epoch and training details for each model are provided in Table 2. For SupMoCo schemes, the models were trained with triple precision labelling (low/medium/high labels) with respect to the blur σ value. For WeakCon, the weighting $w_{ij}$ was found by calculating the Euclidean distance between the normalised kernel widths (σ) of the query and negative samples (as proposed in [25]). The number of training epochs for each encoder was selected qualitatively, based on when the contrastive loss started to plateau, and after degradation clustering could be qualitatively observed on the validation set. Training the encoders beyond this point seemed to provide little to no performance gain, as is shown in Table 3.

We used the trained encoders to generate prediction vectors for the entirety of the BSDS100, Manga109 and Urban100 testing datasets, with each image degraded with three different kernel widths (927 images in total), and then applied t-SNE reduction for each set of outputs. We also fed this test set to a DASR pretrained encoder (using the weights provided in [24]) for direct comparison with our own networks. The t-SNE results are presented in Figure 6.

It is immediately apparent that all of the models achieve some level of separation between the three different σ values. However, the semi-supervised methods produce very clear clustering (with just a few outliers) while the MoCo methods generate clusters with less well-defined edges. The influence of the labelling systems clearly produces a very large repulsion effect between the different σ widths, which the unsupervised MoCo system cannot match. Interestingly, there is no discernable distinction between the WeakCon and SupMoCo plots, despite their different modes of action. Additionally, minor modifications to the training process such as swapping the encoder for a larger model (e.g., ResNet) or continuing to train the predictor in tandem with an SR model (SR results in Section 4.4) appear to provide no benefit or even degrade the output clusters.

#### 4.3.2. Regression Analysis

For our iterative and regression models, the output prediction is much simpler to interpret. Direct σ and PCA kernel estimates can be immediately compared with the actual value. We trained a variety of iterative DAN models, using RCAN as our base SR model for consistency. Several separate RCAN-DAN models were implemented; one specifically predicting the σ and others predicting a 10-element PCA representation of each kernel. We also trained two DANv1 models (predicting PCA kernels) from scratch for comparison: one using a fixed learning rate of $2\times 10^{-4}$ (matching the original implementation in [23]) and one using our own cosine annealing scheduler with a restart value of 250,000 (matching our other models). We compared the prediction capabilities of our models, and a number of pretrained checkpoints from the literature, on our testing sets (blurred with multiple values of σ). The pretrained models for IKC, DANv1 and DANv2 were extracted from their respective official code repositories. These pretrained models were also trained on DIV2K/Flickr2K images, but training degradations were generated online (with σ in the range [0.2,4]), which should result in superior performance. Figure 7A shows the prediction error of the direct regression models that were trained (both contrastive and iterative models). The results clearly show that the DAN predictor is the strongest of those tested, with errors below 0.05 in some cases (representing an error of less than 2.5%). The contrastive/regression methods, while producing respectable results in select scenarios, seem to suffer across most of the distribution tested. In both types of models, the error seems to increase when the width is at its lower range. We hypothesise that, at this point, it is difficult to distinguish between σ of 0.2–0.4, given that the corresponding kernels are quite small.

Figure 7B shows the results of the PCA prediction models. The plot shows that our RCAN-DAN models achieve very similar prediction performance to the pretrained DANs. What makes this result remarkable is the fact that our models were trained for much less time than the pretrained DAN models, both of which were trained for ≈7000 epochs. Training DANv1 from scratch for the same amount of time as our models (1000 epochs) shows that the prediction performance at this point is markedly worse. It is clear that the larger and more capable RCAN model is helping boost the σ prediction performance significantly. On the other hand, the pretrained IKC model is significantly outclassed by all DAN models in almost all scenarios. It is also worth noting that the prediction of kernels at the lower end of the spectrum suffers from increased error, across the board.

### 4.4. Blind SR on Simple Pipeline

The real test for our combined SR and predictor models is the blind SR performance. Table 3 presents the blind SR PSNR results of all the models considered on the test sets under various levels of blur σ. SSIM results are also provided in the Appendix A. Figure 8 and Figure 9 further complement these results, with a bar chart comparison of key models and a closer look at the SR performance across various levels of of σ, respectively.

With reference to the model categories highlighted in Table 3, we make the following observations:**Pretrained models**: We provide the results for the pretrained models evaluated in Figure 7B, along with the results for the pretrained DASR [24], a blind SR model with a MoCo-based contrastive encoder. The DAN models have the best results in most cases (with DANv2 having the best performance overall). For IKC, another iterative model, we present two sets of metrics: IKC (pretrained-best-iter) shows the results obtained when selecting the best image from all SR output iterations (7 in total), as is the implementation in the official IKC codebase. IKC (pretrained-last-iter) shows the results obtained when selecting the image from the last iteration (as is done for the DAN models). The former method produces the best results (even surpassing DAN in some cases), but cannot be applied in true blind scenarios where a reference HR image is not available.**Non-blind models**: The non-blind models fed with the true blur kernel width achieve the best performance of all the models studied. This is true both for RCAN and HAN, with HAN having a slight edge overall. The wide margin over all other blind models clearly shows that significantly improved performance is possible if the degradation prediction system can be improved. We also trained and tested a model (RCAN-MA (noisy sigma)) which was provided with the normalised σ values corrupted by noise (mean 0, standard deviation 0.1). This error level is slightly higher than that of the DAN models tested ( Figure 7A), allowing this model to act as a performance reference for our estimation methods.**RCAN-DAN models**: As observed in Figure 7, the DAN models that were trained from scratch are significantly worse than the RCAN models, including the fully non-blind RCAN, across all datasets and σ values, as shown in Figure 8 and Figure 9 respectively. The RCAN-DAN models show a consistent performance boost over RCAN across the board. As noted earlier in Section 4.2, predicting PCA-reduced kernels appears to provide no advantage over directly predicting the kernel width.**RCAN-Contrastive models**: For the contrastive models, the results are much less clear-cut. The different contrastive blind models exhibit superior performance to RCAN under most conditions (except for Urban100), but none of the algorithms tested (MoCo, SupMoCo, WeakCon and direct regression) seem to provide any particular advantage over each other. The encoder trained with combined regression and SupMoCo appears to provide a slight boost over the other techniques (Figure 8), but this is not consistent across the datasets and σ values analysed. This is a surprising result, given that the clear clusters formed by SupMoCo and WeakCon (as shown in Figure 6) would have been expected to improve the encoders’ predictive power. We hypothesise that the encoded representation is difficult for even deep learning models to interpret, and a clear-cut route from the encoded vector to the actual blur σ may be difficult to produce. We also observe that both the RCAN-DAN and RCAN-SupMoCo models clearly surpass the noisy sigma non-blind RCAN model on datasets with medium and high σ, while they perform slightly worse on datasets with low σ. This matches the results in Figure 7, where it is clear that the performance of all predictors suffer when σ is low.**HAN models**: Upgraded HAN models appear to follow similar trends as RCAN models. The inclusion of DAN provides a clear boost in performance, but this time the inclusion of the SupMoCo-regression predictor seems to only boost performance when σ is high.**Extensions**: We also trained a RCAN-DAN model where we pre-initialised the predictor with that from the pretrained DANv1 model. The minor improvements indicate that, for the most part, the predictor is achieving similar prediction accuracy to that of the pretrained models (as is also indicated in Figure 7). We also extended the training of the baseline RCAN and the RCAN-SupMoCo-regression model to 2200 epochs. The expanded training continues to improve performance and, perhaps crucially, the contrastive model continues to show a margin of improvement over the baseline RCAN model. In fact, this extended model starts to achieve similar or better performance than the pretrained DAN models (shown both in Table 3 and Figure 9). This is achieved with a significantly shorter training time (2200 vs. ≈7000 epochs) and a fixed set of degradations, indicating that our models would surpass the performance of the pretrained DAN models if trained with the same conditions.

Additionally, we implemented, trained and tested the Real-ESRGAN and ELAN models with the addition of the MA metadata insertion block (with the same hyperparameters as presented in Section 4.1). The testing results are available in the Appendix A (Appendix A containing Real-ESRGAN LPIPS results, and Appendix A containing the PSNR and SSIM results for ELAN, respectively). For Real-ESRGAN, the addition of the true metadata (non-blind) makes a clear improvement over the base model. We also observed a consistent improvement in performance across datasets and σ values for the DAN upgraded model. However, attaching the best performing SupMoCo encoder provided no clear advantage. We hypothesise that the Real-ESRGAN model is more sensitive to the accuracy of the kernel prediction, and thus sees limited benefit from the less accurate contrastive encoder (as we have shown for the DAN vs. contrastive methods (Figure 7)).

For ELAN, the baseline model is very weak, and is actually surpassed by Lanczos upsampling in one case (both in terms of PSNR and SSIM). The addition of the true metadata only appeared to help when MA was distributed through the whole network, upon which it increased the performance of the network massively (>3 dB in some cases). It is clear that ELAN does not perform well on these blurred datasets (ELAN was originally tested only on bicubically downsampled datasets). However, MA still appears to be able to significantly improve the model’s performance under the right conditions. Further investigation is required to first adapt ELAN for such degraded datasets before attempting to use this model as part of our blind framework.

### 4.5. Complex Degradation Prediction

For our extended analysis on more realistic degradations, we trained three contrastive encoders (MoCo, SupMoCo and WeakCon) and one RCAN-DAN model on the complex pipeline dataset (Section 4.1). Given the large quantity of degradations, we devised a number of testing scenarios, each applied on the combined images of BSDS100, Manga109 and Urban100 (309 images total). The scenarios we selected are detailed in Table 4. We will refer to these testing sets for the rest of this analysis. We evaluated the prediction capabilities of the contrastive and iterative models separately. We purposefully limited the testing blur kernel shapes to isotropic/anisotropic Gaussians to simplify analysis.

#### 4.5.1. Contrastive Learning

For each of the contrastive algorithms, we trained an encoder (all with the same architecture as used for the simple pipeline) with the following protocol:We first pre-trained the encoder with an online pipeline of noise (same parameters as the full complex pipeline, but with an equal probability to select grey or colour noise) and bicubic downsampling. We found that this pre-training helps reduce loss stagnation for the SupMoCo encoder, so we applied this to all encoders. The SupMoCo encoder was trained with double precision at this stage. We used 3 positive patches for SupMoCo and 1 positive patch for both MoCo and WeakCon.After 1099 epochs, we started training the encoder on the full online complex pipeline (Section 4.1). The SupMoCo encoder was switched to triple precision from this point onwards.We stopped all encoders after 2001 total epochs, and evaluated them at this checkpoint.For SupMoCo, the decision tree in Section 3.5.2 was used to assign class labels. For WeakCon, $w_{ij}$ was computed as the Euclidean distance between query/negative sample vectors containing: the vertical and horizontal blur σ, the Gaussian/Poisson sigma/scale, respectively, and the JPEG/JM H.264 quality factor/QPI, respectively, (6 elements total). All values were normalised to [0,1] prior to computation.

As with the simple pipeline, contrastive encodings are not directly interpretable and so we analysed the clustering capabilities of each encoder through t-SNE visualizations. We evaluated each encoder on the full testing scenario (*Iso/Aniso + Gaussian/Poisson + JPEG/JM* in Table 4), and applied t-SNE independently for each model. The results are shown in Figure 10.

It is evident from the t-SNE plots that the clustering of the dataset is now significantly more complex than that observed in Figure 6. However, all three encoders appear to have successfully learnt how to distinguish between the two compression types and are also mostly successful when clustering the four types of noise (MoCo is slightly weaker for grey noise). In the Appendix A, we also show that the encoders are capable of separating different intensities of both compression and noise, albeit with less separation of the two noise types.

For blurring, the separation between isotropic and anisotropic kernels is much less logical. It appears that each encoder was attempting to form sub-clusters for each type of kernel in some cases (in particular for SupMoCo) but the separation is significantly less clear cut than that obtained in Figure 6. Further analysis would be required to decipher whether clustering is weak simply due to the difficulty of the exercise, or whether clustering is being mostly influenced by the other degradations considered in the pipeline.

As observed with the simple pipeline, it is again apparent that the different methods of semi-supervision seem to be converging to similar results. This is also in spite of the fact that WeakCon was supplied with only 6 degradation elements while SupMoCo was supplied with the full degradation metadata through its class system. Further investigation into their learning process could reveal further insight into the effects of each algorithm.

#### 4.5.2. Iterative Parameter Regression

The RCAN-DAN model was trained on the complex pipeline dataset with identical hyperparameters to that of the simple pipeline. For degradation prediction, we set the DAN model to predict a vector with the following elements (15 total):Individual elements for the following blur parameters: vertical and horizontal σ, rotation, individual β for generalised Gaussian and plateau kernels and the sinc cutoff frequency. Whenever one of these elements was unused (e.g., cutoff frequency for Gaussian kernels), this was set to 0. All elements were normalised to [0,1] according to their respective ranges (Section 4.1).Four boolean (0 or 1) elements categorising whether the kernel shape was:−Isotropic or anisotropic−Generalised−Plateau-type−SincIndividual elements for the Gaussian sigma and Poisson scale (both normalised to [0,1]).A boolean indicating whether the noise was colour or grey type.Individual elements for the JM H.264 QPI and JPEG quality factor (both normalised to [0,1]).

We tested the prediction accuracy by evaluating the model on a number of our testing scenarios, and then quantifying the degradation prediction error. The results are shown in Table 5. As observed with the contrastive models, blur kernel parameter prediction accuracy is extremely low, even when no other degradations are present. On the other hand, both noise and compression prediction are significantly better, with sub 0.1 error in all cases, even when all degradations are present. We hypothesise that since blur kernels are introduced as the first degradation in the pipeline, most of the blurring information could be masked when noise addition and compression have been applied.

To the best of our knowledge, we are the first to present fully explicit blind degradation prediction on this complex pipeline. We hope that the prediction results achieved in this analysis can act as a baseline from which further advances and improvements can be made.

### 4.6. Blind SR on Complex Pipeline

For blind SR on the complex pipeline, we focus on just the RCAN and RCAN upgraded models to simplify analysis. We use a single MA block to insert metadata into the SR core in all cases apart from one, where we distribute MA throughout RCAN. We also trained a number of non-blind models (fed with different quantities of the correct metadata) as comparison points. PSNR SR results comparing the baseline RCAN to the blind models are provided in Table 6 (Appendix A provides the SSIM results).

We make the following observations on these results:**Compression- and noise-only scenarios:** In these scenarios, the RCAN-DAN model shows clear improvement over all other baseline and contrastive encoders (apart from some cases on Manga109). Improvement is most significant in the compression scenarios.**Blur-only scenarios:** Since the blurring scenarios are very similar or identical to the simple pipeline, the models from Table 3 (RCAN is also shown in Table 6) are significantly stronger. The DAN model overtakes the baseline in some cases, but is very inconsistent.**Multiple combinations:** In the multiple degradation scenarios, the DAN model consistently overtakes the baselines, but PSNR/SSIM increases are minimal.

For all scenarios, there are a number of other surprising results. The contrastive methods appear to be providing no benefit to SR performance, in almost every case. Furthermore, the non-blind models are often overtaken by the DAN model in certain scenarios, and the amount of metadata available to the non-blind models does not appear to correlate with the final SR performance. It is clear that the metadata we have available for these degradations are having a much lesser impact on SR performance than on the simple pipeline. Since the contrastive encoders have shown to be slightly weaker than DAN in the simple pipeline case (Figure 7), it is clear that their limited prediction accuracy is also limiting potential gains in SR performance on this pipeline. This dataset is significantly more difficult than the simple case, not just due to the increased amount of degradations, but also as the models appear less receptive to the insertion of metadata. We again hope that these results can act as a baseline for further exploration into complex blind SR.

### 4.7. Blind SR on Real LR Images

As a final test to compare models from both pipelines, we ran a select number of models on real-world images from RealSRSet [19]. These results are shown in Figure 11, with an additional image provided in the Appendix A. This qualitative inspection clearly show that models trained on the complex pipeline are significantly better at dealing with real-world degradations than simple pipeline models. Figure 11 shows that the complex pipeline models can remove compression artefacts, sharpen images and smoothen noise. In particular, the dog image shows that RCAN-DAN can deal with noise more effectively than the baseline RCAN. The simple pipeline model results are all very similar to each other, as none of them are capable of dealing with degradations other than isotropic blurring.

### 4.8. Results Summary

Given the large quantity of analyses conducted, we provide a brief summary of the most significant results obtained in each section here:In Section 4.2, we show that all of the metadata insertion mechanisms tested provide roughly the same SR performance boost when feeding a large network such as RCAN with non-blind blurring metadata. Furthermore, adding repeated blocks through the network provides little to no benefit. Given this result, we propose MA as our metadata insertion block of choice, as it provides identical SR performance as the other options considered, with very low complexity. Other metadata blocks could prove optimal in other scenarios (such as other degradations or with other networks), which would require further systematic investigation to determine.Section 4.3 provides a comparison of the prediction performance of the different algorithms considered on the simple blur pipeline. The contrastive algorithms clearly cluster images by the applied blur kernel width, with the semi-supervised algorithms providing the most well-defined separation between different values. The regression and iterative mechanisms are capable of explicitly predicting the blur kernel width with high accuracy, except at the lower extreme. Our prediction mechanisms combined with RCAN match the performance of the pretrained DAN models with significantly less training time.Section 4.4 compares the testing results of blind models built with our framework with baseline models from the literature. Each prediction mechanism considered elevates RCAN’s SR performance above its baseline value, for both PSNR and SSIM. In particular, the iterative mechanism provided the largest performance boost. For more complex models such as HAN and Real-ESRGAN, contrastive methods provide less benefit, but the iterative mechanism still shows clear improvements. Our models significantly overtake the SOTA blind DAN network when trained for the same length of time. In addition, our models approach or surpass the performance of the pretrained DANv1 and DANv2 checkpoints provided by their authors, which were trained for a significantly longer period of time.In Section 4.5 and Section 4.6, we modify our prediction mechanisms to deal with a more complex pipeline of blurring, noise and compression, and attach these to the RCAN network. We show that the contrastive predictors can reliably cluster compression and noise, but blur kernel clustering is significantly weaker. Similarly, the iterative predictors are highly accurate when predicting compression/noise parameters, but are much less reliable for blur parameters.When testing their SR performance, the contrastive encoders seem to provide little to no benefit to RCAN’s performance. On the other hand, the DAN models reliably improve the baseline performance across various scenarios, albeit with limited improvements when all degradations are present at once. We anticipate that performance can be significantly improved with further advances to the prediction mechanisms and consider our results as a baseline for further exploration.Section 4.7 showcases the results of our models when applied to real-world LR images. Our complex pipeline models produce significantly better results than the pretrained DAN models and are capable of reversing noise, compression and blurring in various scenarios.

## 5. Conclusions

In this work, a framework for combining degradation prediction systems with any SR network was proposed. By using a single metadata insertion block to influence the feature maps of a convolutional layer, a degradation vector from a prediction model can, in many cases, be used to improve the performance of the SR network. This premise was tested by implementing several contrastive and iterative degradation prediction mechanisms and coupling them with high-performing SR architectures. When tested on a dataset of images that were degraded by Gaussian blurring and downsampling, we show that our blind mechanisms achieve at least the same (or better) blur σ prediction accuracy as the original methods, but with significantly less training time. Moreover, both blind degradation performance (in combined training cases, such as with DAN) and SR performance are substantially improved through the use of larger and stronger networks such as RCAN [8] or HAN [10]. Our results show that our hybrid models surpass the performance of the baseline non-blind and blind models under the same conditions. Other SR architecture categories such as the SOTA perceptual-loss based Real-ESRGAN [16] and the transformer-based ELAN architecture [13] also work within our framework, but the performance of these methods is more sensitive to the accuracy of the degradation prediction and the dataset used for training. We show that this premise also holds true for blind SR of a more complex pipeline involving various blurring, noise injection, and compression operations.

Our framework should enable blind SR research to be significantly expedited, since researchers can now focus their efforts on their degradation prediction mechanisms, rather than on deriving a custom SR architecture for each new method. There are various future avenues that could be explored to further assess the applications of our framework. Apart from investigating new combinations of blind prediction, metadata insertion and SR architectures, our framework could also be applied to new types of metadata. For example, blind prediction systems could be replaced with image classification systems. This will provide SR architecture with details on the image content (e.g., facial features for face SR [36]). Furthermore, the framework can be extended to video SR [91] where additional sources of metadata are available, such as the number of frames to be used in the super-resolution of a given frame as well as other details on the compression scheme, such as P- and B-frames (in addition to I-frames as considered in this work). 

## Figures and Tables

**Figure 1 sensors-23-00419-f001:**
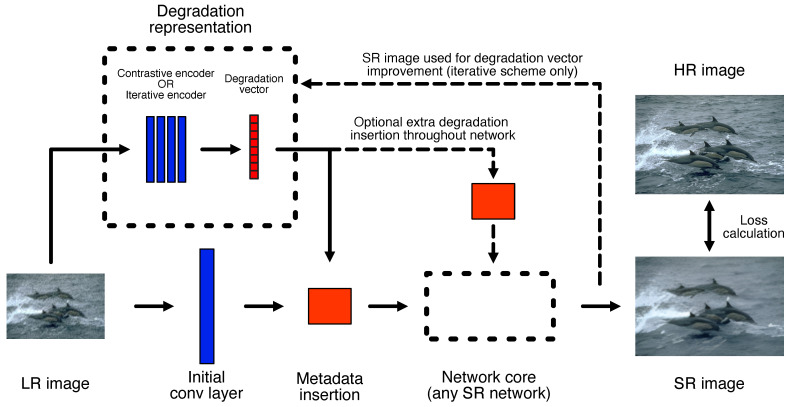
Proposed framework for combining blind degradation systems and SR models. The metadata insertion block acts as the bridge between the two systems, allowing the SR model to exploit the predicted degradation to improve its performance. Depending on the blind predictor mechanism chosen, the SR image can be fed back into the predictor to help improve its accuracy.

**Figure 2 sensors-23-00419-f002:**
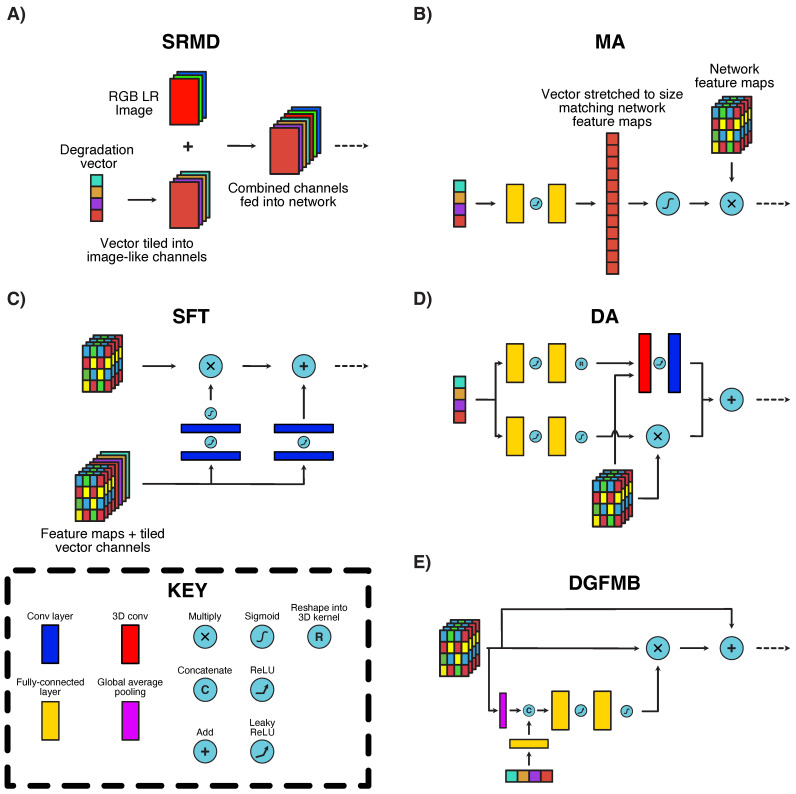
Metadata insertion mechanisms investigated in this paper. (**A**) SRMD-style metadata insertion. (**B**) Meta-Attention block. (**C**) SFT block. (**D**) Degradation-Aware block. (**E**) Degradation-Guided Feature Modulation Block. The end result of each mechanism is the trainable modulation of CNN feature maps using the information present in a provided vector. Each mechanism varies substantially in complexity, positioning and the components involved.

**Figure 3 sensors-23-00419-f003:**
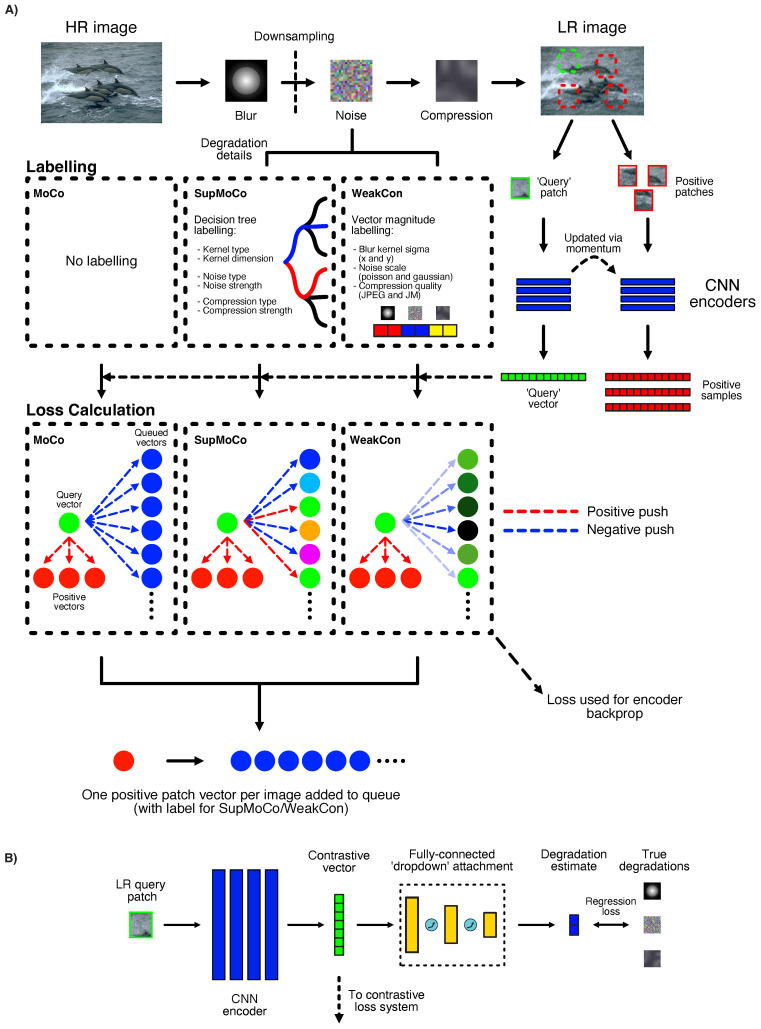
Contrastive learning for SR. (**A**) The three contrastive learning mechanisms considered in this work. While MoCo is the simplest and the only fully unsupervised method, SupMoCo and WeakCon provide more targeted learning at the expense of requiring user-defined labelling systems. (**B**) Direct parameter regression can also be used to add an additional supervised element to the encoder training process.

**Figure 4 sensors-23-00419-f004:**
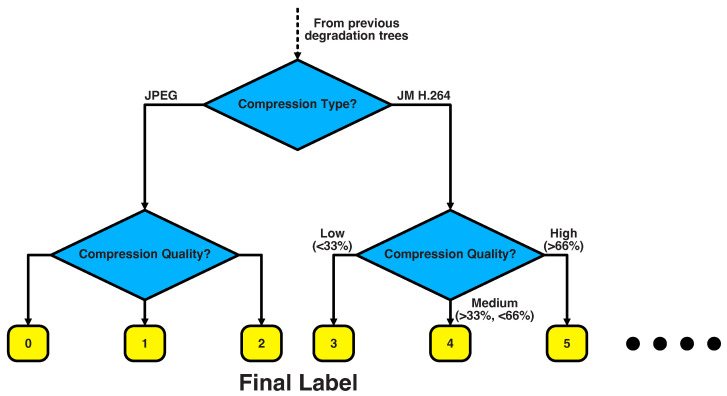
An example of the labelling decision tree, applied to compression degradations. The final label is used to direct the contrastive loss in a SupMoCo system. Other degradation types can be linked to this tree, further diversifying the labels available.

**Figure 5 sensors-23-00419-f005:**
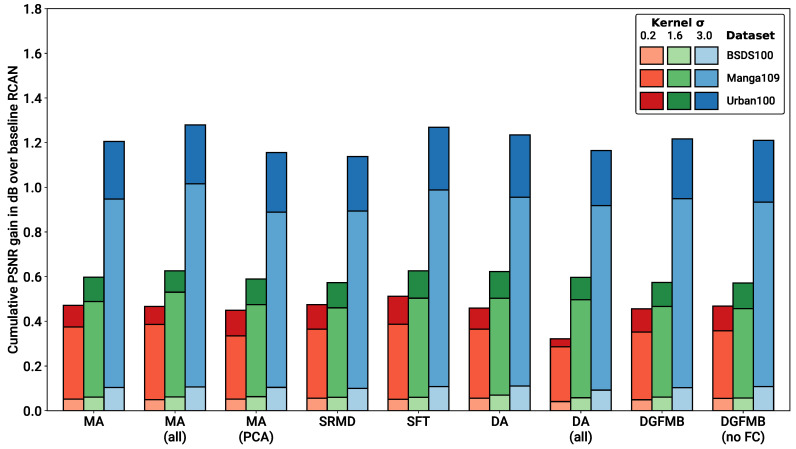
Bar graph showing improvement in PSNR over baseline RCAN for each metadata insertion block. PSNR improvements are stacked on each other for each specific σ to show cumulative PSNR gain across datasets.

**Figure 6 sensors-23-00419-f006:**
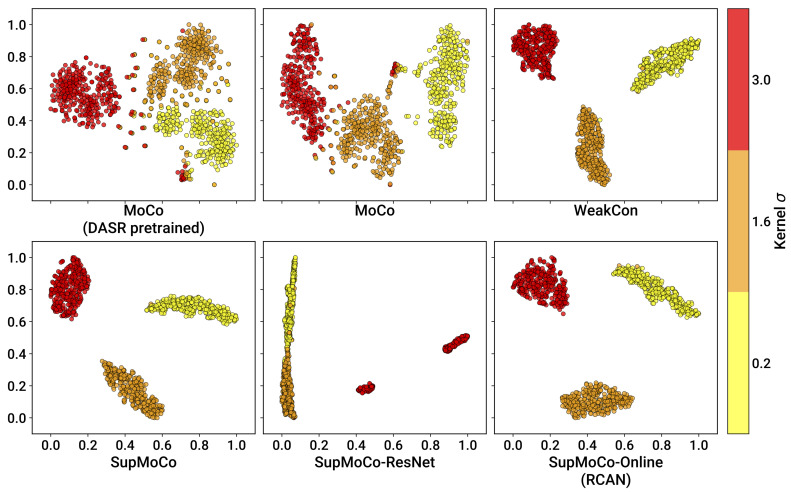
t-SNE plots (perplexity value of 40) showing the separation power of the different contrastive learning algorithms considered. Each dimension was independently normalised in the range [0,1] after computing the t-SNE results on the encoded vectors generated on our test set of 927 LR images.

**Figure 7 sensors-23-00419-f007:**
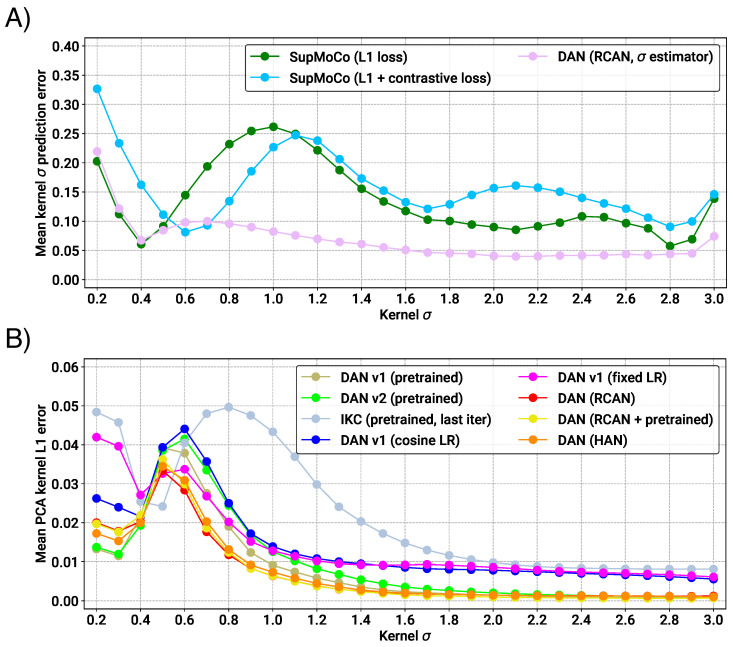
Prediction capabilities of regression and iterative-based encoders on the BSDS100, Manga109 and Urban100 testing sets (total of 309 images for each σ value). (**A**) Plot showing the relation between the average kernel prediction error and the actual kernel width for direct regression models. (**B**) Plot showing the relation between average prediction error (in PCA-space) and the actual σ for iterative models.

**Figure 8 sensors-23-00419-f008:**
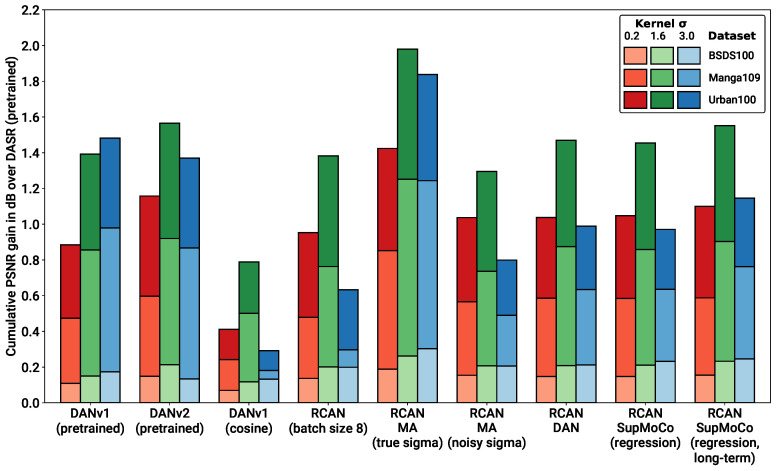
Bar graph showing improvement in PSNR over the pretrained DASR, for key models from Table 3. PSNR improvements are stacked on each other for each specific blur kernel σ to show cumulative PSNR gain across datasets. Our hybrid models can match and even surpass the pretrained DAN models, despite the large disparity in training time.

**Figure 9 sensors-23-00419-f009:**
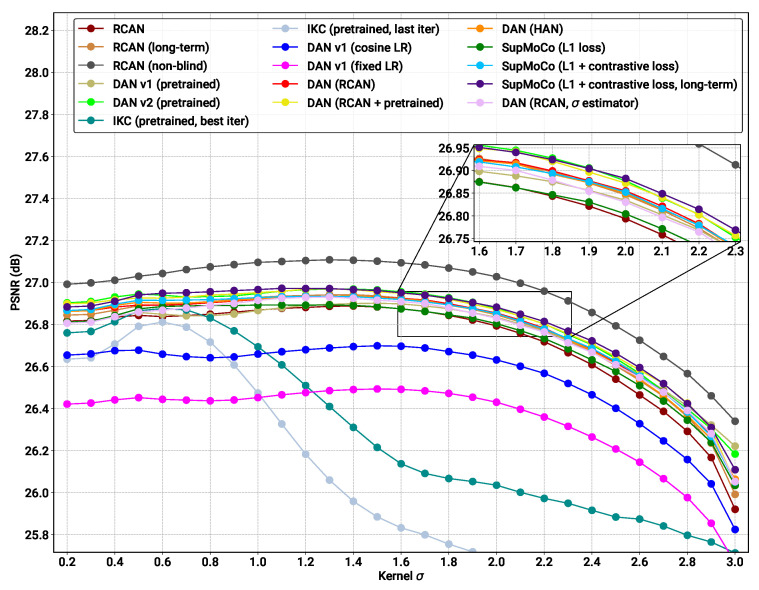
Plot showing the relationship of SR performance (as measured using PSNR) with the level of blurring within an image. PSNR was measured by running each model on the BSDS100, Manga109 and Urban100 datasets (309 images) degraded with each specified σ value. ‘L1 loss’ refers to the regression component added to our contrastive encoders (Figure 3). All models appear to show degraded performance with higher levels of blurring, indicating that none of the models analysed are capable of fully removing blurring effects from the final image. The non-blind model outperforms all other models, indicating further improvements are possible with better degradation prediction.

**Figure 10 sensors-23-00419-f010:**
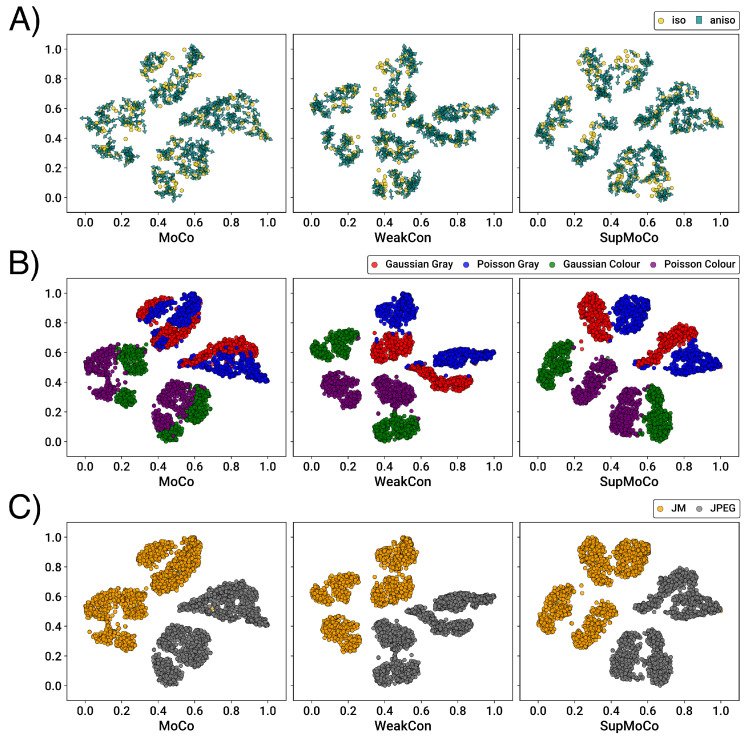
t-SNE plots (perplexity value of 40) showing the separation power of the different contrastive learning algorithms considered on the complex pipeline. All models were evaluated on the *Iso/Aniso + Gaussian/Poisson + JPEG/JM* testing scenario having 4944 images (ref. Table 4). Each dimension was independently normalised in the range [0,1] after computing thet-SNE results. The data used by each method is identical across rows, with different colours pertaining to the degradation considered: (**A**) t-SNE plots with each point labelled according to the blur kernel applied. Only 560 images (randomly selected) are shown for each plot, to reduce cluttering. Arrows indicate the rotation of the anistropic kernels. (**B**) t-SNE plots with each point coloured according to the noise injected. (**C**) t-SNE plots with each point coloured according to the compression applied.

**Figure 11 sensors-23-00419-f011:**
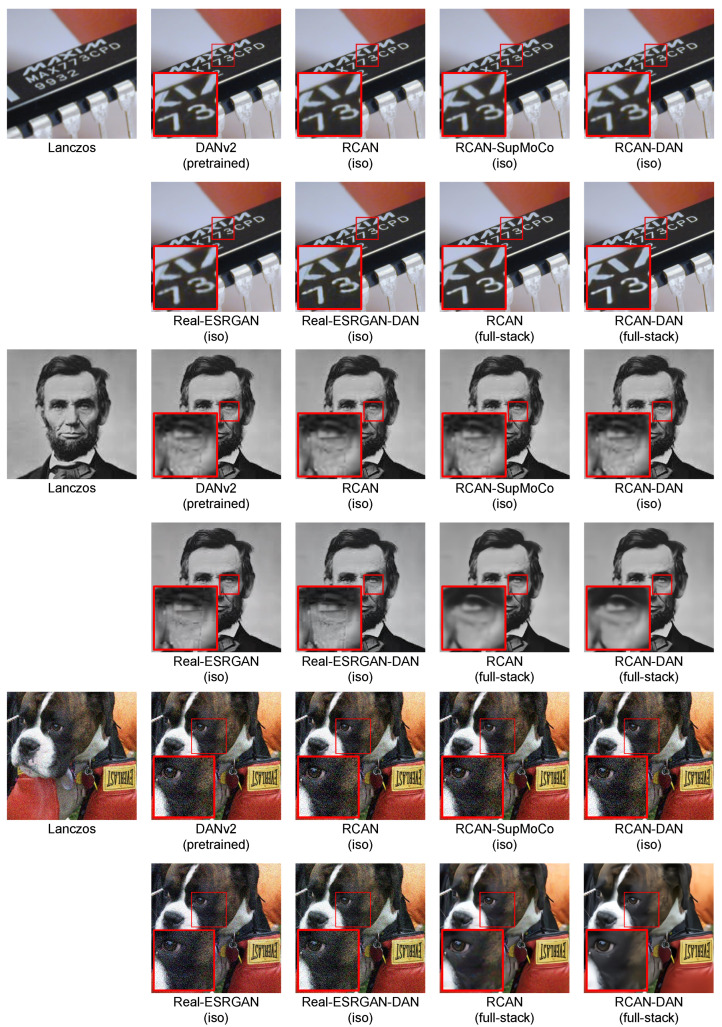
Comparison of the SR results for various models, on images from RealSRSet [19]. All simple pipeline models (marked as ‘iso’) and the pretrained DAN model are incapable of dealing with degradations such as noise or compression. The complex pipeline models (marked as‘full-stack’) produce significantly improved results. These models can sharpen details (first image), remove compression artefacts (second image) and smooth out noise (third image).

**Table 1 sensors-23-00419-t001:** PSNR (dB) SR results on simple pipeline comparing metadata insertion blocks. The labels ‘low’, ‘med’ and ‘high’ refer to σ values of 0.2, 1.6, and 3.0, respectively. Models in the ‘Non-Blind’ category are all RCAN models, upgraded with one instance of the indicated metadata insertion block, unless stated otherwise. ‘MA (all)’ and ‘DA (all)’ refer to RCAN with 200 individual MA or DA blocks inserted throughout the network. ‘MA (PCA)’ refers to RCAN with a single MA block provided with a 10-element PCA-reduced vector of the blur kernel applied. ‘DGFMB (no FC)’ refers to RCAN with a DGFMB layer where the metadata input is not passed through a fully-connected layer prior to concatenation (refer to Figure 2E). The best result for each set is shown in red, while the second-best result is shown in blue.

Model	Set5			Set14			BSDS100			Manga109			Urban100		
	Low	Med	High	Low	Med	High	Low	Med	High	Low	Med	High	Low	Med	High
**Baselines**															
Bicubic	27.084	25.857	23.867	24.532	23.695	22.286	24.647	23.998	22.910	23.608	22.564	20.932	21.805	21.104	19.944
Lanczos	27.462	26.210	24.039	24.760	23.925	22.409	24.811	24.173	23.007	23.923	22.850	21.071	21.989	21.293	20.046
RCAN	30.675	30.484	29.635	27.007	27.003	26.149	26.337	26.379	25.898	29.206	29.406	27.956	24.962	24.899	23.960
**Non-Blind**															
MA	30.956	30.973	29.880	27.091	27.094	26.383	26.389	26.440	26.002	29.529	29.834	28.799	25.059	25.008	24.218
MA (all)	30.926	30.911	29.911	27.057	27.066	26.373	26.386	26.441	26.005	29.542	29.874	28.865	25.043	24.995	24.224
MA (PCA)	30.947	30.942	29.873	27.052	27.038	26.382	26.388	26.442	26.003	29.489	29.817	28.740	25.077	25.014	24.227
SRMD	30.946	30.960	29.906	27.086	27.072	26.358	26.393	26.439	25.998	29.515	29.806	28.750	25.072	25.012	24.204
SFT	30.960	30.958	29.972	27.065	27.066	26.386	26.388	26.439	26.006	29.541	29.849	28.835	25.088	25.022	24.241
DA	30.930	30.934	29.929	27.088	27.093	26.389	26.392	26.449	26.009	29.515	29.839	28.801	25.056	25.019	24.239
DA (all)	30.956	30.958	29.888	27.033	27.044	26.358	26.378	26.437	25.990	29.451	29.844	28.781	24.998	24.999	24.207
DGFMB	30.969	30.955	29.891	27.068	27.060	26.347	26.386	26.440	26.001	29.508	29.811	28.802	25.067	25.007	24.228
DGFMB (no FC)	30.985	30.941	29.909	27.062	27.077	26.388	26.392	26.436	26.006	29.508	29.806	28.781	25.073	25.014	24.237

**Table 2 sensors-23-00419-t002:** Contrastive encoders considered in our ‘simple pipeline’ analysis. The ‘positive patches per query patch’ column refers to how many patches are produced per iteration to act as positive samples, as stipulated in Equations (Equation 6)–(Equation 8).

Model	Positive Patches per Query Patch	Epoch Selected	Loss
MoCo	1	2104	Equation (Equation 6)
SupMoCo	3	542	Equation (Equation 7)
WeakCon	1	567	Equation (Equation 8)
SupMoCo (regression)	N/A	376	L1 loss
SupMoCo (contrastive + regression)	3	115	Equation (Equation 7) & L1 loss
SupMoCo (ResNet)	3	326	Equation (Equation 7)

**Table 3 sensors-23-00419-t003:** PSNR (dB) SR results on the simple pipeline comparing blind SR methods. The labels ‘low’, ‘med’ and ‘high’ refer to σ values of 0.2, 1.6, and 3.0, respectively. For RCAN/HAN models with metadata, this is inserted using one MA block at the front of the network. ‘noisy sigma’ refers to the use of normalised σ metadata that has been corrupted with Gaussian noise (mean 0, standard deviation of 0.1) for both training and testing. ‘Long term’ refers to models that have been screened after training for 2200 epochs, rather than the standard 1000. The best result for each set is shown in red, while the second-best result is shown in blue.

Model	Set5			Set14			BSDS100			Manga109			Urban100		
	Low	Med	High	Low	Med	High	Low	Med	High	Low	Med	High	Low	Med	High
**Classical**															
Bicubic	27.084	25.857	23.867	24.532	23.695	22.286	24.647	23.998	22.910	23.608	22.564	20.932	21.805	21.104	19.944
Lanczos	27.462	26.210	24.039	24.760	23.925	22.409	24.811	24.173	23.007	23.923	22.850	21.071	21.989	21.293	20.046
**Pretrained**															
IKC (pretrained-best-iter)	30.828	30.494	30.049	26.981	26.729	26.603	26.303	26.195	25.890	29.210	27.926	27.416	24.768	24.291	23.834
IKC (pretrained-last-iter)	30.662	30.057	29.812	26.904	26.628	26.181	26.238	25.998	25.659	29.034	27.489	26.617	24.633	24.010	23.566
DASR (pretrained)	30.545	30.463	29.701	26.829	26.701	26.060	26.200	26.178	25.699	28.865	28.844	27.858	24.487	24.280	23.624
DANv1 (pretrained)	30.807	30.739	30.049	26.983	26.925	26.360	26.309	26.329	25.873	29.230	29.549	28.664	24.897	24.817	24.127
DANv2 (pretrained)	30.850	30.881	30.042	27.033	26.999	26.314	26.349	26.392	25.832	29.313	29.550	28.592	25.048	24.926	24.127
**Non-Blind**															
RCAN-MA (true sigma)	30.956	30.973	29.880	27.091	27.094	26.383	26.389	26.440	26.002	29.529	29.834	28.799	25.059	25.008	24.218
HAN-MA (true sigma)	30.940	30.905	29.834	27.029	27.067	26.387	26.391	26.444	26.003	29.543	29.844	28.832	25.089	25.021	24.234
RCAN-MA (noisy sigma)	30.838	30.700	29.780	27.023	26.966	26.168	26.354	26.386	25.905	29.276	29.373	28.142	24.959	24.839	23.933
**RCAN/DAN**															
DANv1	30.432	30.387	29.436	26.773	26.754	26.022	26.178	26.212	25.762	28.709	28.937	27.656	24.378	24.326	23.583
DANv1 (cosine)	30.627	30.466	29.537	26.858	26.855	26.101	26.270	26.296	25.831	29.037	29.227	27.908	24.657	24.568	23.735
RCAN (batch size 4)	30.686	30.538	29.646	26.957	26.966	26.198	26.324	26.376	25.907	29.172	29.395	28.160	24.893	24.846	23.943
RCAN-DAN	30.813	30.627	29.741	26.997	26.996	26.208	26.348	26.387	25.912	29.303	29.509	28.280	24.940	24.876	23.979
RCAN-DAN (sigma)	30.747	30.666	29.782	27.013	27.014	26.235	26.340	26.391	25.924	29.284	29.462	28.275	24.798	24.874	23.958
**HAN/DAN**															
HAN (batch size 4)	30.779	30.620	29.572	26.998	27.002	26.174	26.341	26.383	25.897	29.226	29.448	28.013	24.922	24.845	23.877
HAN-DAN	30.817	30.567	29.707	27.021	26.993	26.221	26.356	26.388	25.908	29.289	29.477	28.287	24.943	24.899	23.995
**RCAN/Contrastive**															
RCAN (batch size 8)	30.675	30.484	29.635	27.007	27.003	26.149	26.337	26.379	25.898	29.206	29.406	27.956	24.962	24.899	23.960
RCAN-MoCo	30.870	30.677	29.714	27.032	26.980	26.208	26.338	26.377	25.918	29.210	29.394	28.243	24.919	24.850	23.952
RCAN-SupMoCo	30.819	30.700	29.720	27.019	27.005	26.185	26.350	26.397	25.910	29.257	29.437	28.216	24.958	24.876	23.961
RCAN-regression	30.730	30.618	29.642	26.955	26.986	26.206	26.336	26.384	25.913	29.188	29.421	28.284	24.911	24.819	23.906
RCAN-SupMoCo-regression	30.861	30.785	29.676	27.015	27.031	26.177	26.348	26.389	25.931	29.302	29.491	28.262	24.950	24.877	23.959
RCAN-WeakCon	30.741	30.600	29.720	27.021	27.011	26.224	26.342	26.387	25.924	29.204	29.433	28.189	24.913	24.862	23.955
RCAN-SupMoCo (online)	30.777	30.695	29.771	27.013	26.992	25.931	26.336	26.389	25.915	29.220	29.401	28.189	24.915	24.850	23.890
RCAN-SupMoCo (ResNet)	30.712	30.604	29.658	26.974	26.984	26.103	26.317	26.363	25.869	29.221	29.422	27.266	24.832	24.785	23.704
**HAN/Contrastive**															
HAN (batch size 8)	30.705	30.592	29.593	27.006	26.993	26.142	26.342	26.394	25.910	29.259	29.498	28.053	24.927	24.895	23.902
HAN-SupMoCo-regression	30.734	30.659	29.741	27.005	26.983	26.192	26.343	26.376	25.913	29.195	29.381	28.278	24.926	24.839	23.955
**Extensions**															
RCAN-DAN (pretrained estimator)	30.763	30.612	29.711	27.036	26.988	26.202	26.355	26.394	25.919	29.371	29.564	28.239	24.971	24.888	23.980
RCAN (batch size 8, long-term)	30.736	30.699	29.723	27.011	27.018	26.171	26.343	26.390	25.914	29.230	29.491	28.101	24.962	24.899	23.960
RCAN-SupMoCo-regression (long-term)	30.832	30.640	29.690	27.023	27.019	26.217	26.355	26.411	25.944	29.297	29.514	28.375	24.999	24.929	24.007

**Table 4 sensors-23-00419-t004:** The different testing scenarios considered for the complex analysis. Each scenario was applied on all images of the BSDS100, Manga109 and Urban100 datasets (309 images total). Cases which include noise have double the amount of images (618), since the pipeline was applied twice: once with colour noise, and once with grey noise. The last scenario consists of every possible combination of the degradations considered (16 total combinations with 4944 images in total, including colour and grey noise). For all cases, isotropic blurring was applied with a σ of 2.0, anisotropic blurring was applied with a horizontal σ of 2.0, a vertical σ of 1.0 and a random rotation, Gaussian/Poisson noise were applied with a sigma/scale of 20.0/2.0, respectively, and JPEG/JM H.264 compression were applied with a quality factor/QPI of 60/30, respectively. All scenarios also included ×4 bicubic downsampling inserted at the appropriate point (following the sequence in Equation (Equation 3)).

Test Scenario	Blurring	Noise	Compression	Total Images
JPEG	N/A	N/A	JPEG	309
JM	N/A	N/A	JM H.264	309
Poisson	N/A	Poisson	N/A	618
Gaussian	N/A	Gaussian	N/A	618
Iso	Isotropic	N/A	N/A	309
Aniso	Anisotropic	N/A	N/A	309
Iso + Gaussian	Isotropic	Gaussian	N/A	618
Gaussian + JPEG	N/A	Gaussian	JPEG	618
Iso + Gaussian + JPEG	Isotropic	Gaussian	JPEG	618
Aniso + Poisson + JM	Anisotropic	Poisson	JM H.264	618
Iso/Aniso + Gaussian/Poisson + JPEG/JM	Isotropic & Anisotropic	Gaussian & Poisson	JPEG & JM H.264	4944

**Table 5 sensors-23-00419-t005:** Mean L1 error for the DAN predictor within the RCAN-DAN model trained on the complex pipeline. Prediction accuracy is high for the compression and noise degradations, but weak for blur. Blur parameters incorporate all the blur-specific numerical parameters and the four booleans. Noise parameters incorporate the Gaussian sigma, Poisson scale and colour/grey boolean. Compression parameters incorporate the JPEG quality factor and JM H.264 QPI. The degradation scenarios tested here are described in Table 4.

Test Scenario	Blurring	Noise	Compression
	(σ, kernel type)	(scale, gray/colour type)	(QPI/quality)
**Iso/Aniso**	0.318	N/A	N/A
**Gaussian/Poisson**	N/A	0.042	N/A
**JPEG/JM**	N/A	N/A	0.078
**Iso/Aniso + Gaussian/Poisson + JPEG/JM**	0.317	0.036	0.070

**Table 6 sensors-23-00419-t006:** PSNR (dB) SR results on the complex pipeline comparing blind SR methods. Metadata is inserted into RCAN models using one MA block at the front of the network, apart from models with the ‘all’ suffix where 200 independent MA blocks are inserted throughout the network. The degradation scenarios tested here are described in Table 4. The non-blind models were fed with a vector containing the vertical/horizontal blur σ, the kernel type (7 possible shapes), the Gaussian sigma/Poisson scale, a boolean indicating the addition of grey or colour noise, and the JPEG quality factor/JM H.264 QPI. Non-blind models marked with a degradation (e.g., no blur) have the marked degradation parameters removed from their input vector. All values were normalised to [0,1] apart from the kernel type. The best result for each set is shown in red, while the second-best result is shown in blue.

Dataset	Model	JPEG	JM	Poisson	Gaussian	Iso	Aniso	Iso + Gaussian	Gaussian + JPEG	Iso + Gaussian + JPEG	Aniso + Poisson + JM	Iso/Aniso + Gaussian/Poisson + JPEG/JM
	Bicubic	23.790	23.843	21.714	21.830	23.689	24.014	21.269	21.588	21.185	21.376	21.266
	Lanczos	23.831	23.944	21.435	21.558	23.845	24.185	21.034	21.318	20.969	21.172	21.054
	RCAN (batch size 4)	24.443	24.566	23.847	23.910	25.416	25.492	23.331	23.516	23.023	23.081	23.049
	RCAN (batch size 8)	24.428	24.541	23.868	23.895	25.382	25.443	23.315	23.510	23.026	23.079	23.047
	RCAN (non-blind)	N/A	N/A	N/A	N/A	N/A	N/A	N/A	N/A	23.052	23.007	23.048
	RCAN (non-blind, no blur)	N/A	N/A	N/A	N/A	N/A	N/A	N/A	23.527	23.039	23.019	23.045
	RCAN (non-blind, no noise)	N/A	N/A	N/A	N/A	N/A	N/A	N/A	N/A	22.997	23.081	23.038
	RCAN (non-blind, no compression)	N/A	N/A	N/A	N/A	N/A	N/A	23.346	N/A	23.049	23.041	23.033
	RCAN (MoCo)	24.388	24.548	23.823	23.848	25.246	25.399	23.311	23.505	23.022	23.087	23.048
	RCAN (WeakCon)	24.422	24.532	23.770	23.634	25.297	25.418	23.197	23.510	23.025	23.090	23.051
	RCAN (SupMoCo)	24.390	24.507	23.710	23.751	25.254	25.405	23.237	23.516	23.023	23.087	23.049
	RCAN (SupMoCo, all)	24.412	24.572	23.805	23.873	25.329	25.431	23.298	23.530	23.028	23.088	23.054
	RCAN-DAN	24.447	24.589	23.893	23.920	25.412	25.509	23.343	23.527	23.033	23.092	23.058
**BSDS100**	RCAN (trained on simple pipeline)	23.691	24.059	18.838	18.946	26.335	25.733	18.778	19.053	19.172	18.803	18.955
	Bicubic	22.814	22.983	20.757	21.418	22.084	22.587	20.402	21.170	20.300	20.120	20.201
	Lanczos	22.980	23.225	20.605	21.330	22.329	22.869	20.334	21.079	20.245	20.029	20.126
	RCAN (batch size 4)	25.164	25.419	24.471	24.753	26.387	26.433	23.618	23.987	23.031	23.104	23.060
	RCAN (batch size 8)	25.198	25.561	24.453	24.733	26.264	26.270	23.598	23.982	23.039	23.104	23.062
	RCAN (non-blind)	N/A	N/A	N/A	N/A	N/A	N/A	N/A	N/A	23.140	23.106	23.119
	RCAN (non-blind, no blur)	N/A	N/A	N/A	N/A	N/A	N/A	N/A	24.206	23.087	23.136	23.121
	RCAN (non-blind, no noise)	N/A	N/A	N/A	N/A	N/A	N/A	N/A	N/A	23.099	23.084	23.084
	RCAN (non-blind, no compression)	N/A	N/A	N/A	N/A	N/A	N/A	23.722	N/A	23.146	23.122	23.123
	RCAN (MoCo)	24.961	25.430	24.255	24.509	25.878	26.118	23.599	23.797	23.023	23.096	23.051
	RCAN (WeakCon)	25.115	25.498	23.643	23.838	25.967	26.102	23.130	23.748	23.010	23.083	23.037
	RCAN (SupMoCo)	25.195	25.612	23.921	24.570	25.869	26.114	23.531	24.025	23.023	23.107	23.056
	RCAN (SupMoCo, all)	25.335	25.823	24.329	24.786	25.963	26.161	23.479	24.122	22.992	23.084	23.035
	RCAN-DAN	25.315	25.770	24.369	24.715	26.447	26.431	23.652	24.051	23.082	23.140	23.098
**Manga109**	RCAN (trained on simple pipeline)	23.148	23.954	18.443	19.498	29.329	26.539	18.746	19.340	19.056	17.851	18.406
	Bicubic	21.244	21.353	19.934	20.073	20.772	21.140	19.339	19.864	19.245	19.468	19.345
	Lanczos	21.332	21.496	19.787	19.944	20.939	21.326	19.239	19.725	19.150	19.375	19.248
	RCAN (batch size 4)	22.564	22.854	22.201	22.263	23.130	23.236	21.426	21.814	21.088	21.356	21.214
	RCAN (batch size 8)	22.552	22.840	22.214	22.238	23.115	23.214	21.418	21.816	21.099	21.364	21.221
	RCAN (non-blind)	N/A	N/A	N/A	N/A	N/A	N/A	N/A	N/A	21.165	21.369	21.239
	RCAN (non-blind, no blur)	N/A	N/A	N/A	N/A	N/A	N/A	N/A	21.855	21.105	21.380	21.210
	RCAN (non-blind, no noise)	N/A	N/A	N/A	N/A	N/A	N/A	N/A	N/A	21.128	21.359	21.203
	RCAN (non-blind, no compression)	N/A	N/A	N/A	N/A	N/A	N/A	21.494	N/A	21.159	21.389	21.267
	RCAN (MoCo)	22.559	22.851	22.159	22.182	22.934	23.139	21.398	21.793	21.099	21.378	21.227
	RCAN (WeakCon)	22.554	22.847	22.080	21.805	23.027	23.175	21.108	21.809	21.083	21.372	21.215
	RCAN (SupMoCo)	22.520	22.768	21.983	22.024	22.906	23.116	21.293	21.813	21.089	21.374	21.221
	RCAN (SupMoCo, all)	22.541	22.842	22.117	22.190	23.029	23.165	21.327	21.813	21.063	21.344	21.195
	RCAN-DAN	22.596	22.901	22.273	22.297	23.177	23.294	21.459	21.840	21.125	21.393	21.250
**Urban100**	RCAN (trained on simple pipeline)	21.291	22.217	17.835	18.091	24.717	23.753	17.676	17.938	17.894	17.627	17.727

## Data Availability

All code, data and model weights for the analysis presented in this paper are available here: https://github.com/um-dsrg/RUMpy, accessed on 27 December 2022.

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
