# Peer review of "The Best of Both Worlds: A Framework for Combining Degradation Prediction with High Performance Super-Resolution Networks"

_sensors, 2022, doi:10.3390/s23010419_

Round 1

Reviewer 1 Report

This paper presents a framework for combining any blind SR prediction mechanism with any deep SR network, using a metadata insertion block to insert prediction vectors into SR network feature maps. There are some suggestions as following. 

1.The formulas and models in Section 2.1 should be discussed in Section 3.1. Section 2.1 should add descriptions of the current research status of the Degradation Models to make the article more organized.

2.The captions in all figures and tables should provide a brief summary of the content, and the specific descriptions should be placed in the paragraph descriptions.

Reviewer 2 Report

In this paper, the authors present a framework for combining any blind SR prediction mechanism with any deep SR network, using a metadata insertion block to insert prediction vectors into SR network feature maps. The paper deals with an interesting topic, however,

1. What is the motivation of the proposed method? What is the motivation for the present technique? Please add this detailed description in section 1.

2. In the abstract, the authors indicate that their framework is available at: https://github.com/um-dsrg/RUMpy. But the link is unavailable.

3. The study lacks a clear comparison between the submitted paper and the more relevant literature contributions, which should highlight the main advantages of the current submission.

4. I encourage the authors to have their manuscript proof-edited by a native English speaker to enhance paper presentation levels. 

Reviewer 3 Report

The authors present a Framework for Combining Degradation Prediction with High Performance Super-Resolution Networks. I have a few concerns which should be addressed by the authors.

1. Abstract must be rewritten as it starts with theory and does not exactly properly convey the contributions of the proposed methodology.

2. The article is very lengthy (37 pages). The way it is written is more like a thesis. Authors should put serious and reduce the length of paper to 12-14 pages.

3. Related work section is very long (pages 3-8), reduce it to 1-1.5 pages

4. Similarly, the methodology section is very long, it should be reduced. All of the equations which are taken from the literature should be properly cited.

5. Equations 6-8 highlight the changes (in blue color). However, neither rationale is given nor the original equations are given. Authors should make sure that the rationale they provide should be strong enough to convince the reviewer that the change is a high impact, otherwise, claiming a slight change in the equation as a major breakthrough is not appropriate

6. Authors must include the results computed with original equations (5-8) and the results computed with the proposed modified equations for a fair comparison.

7. DIV2K datasets has 800 images whereas the Flickr2K dataset has 2650 images which are way to less to draw any statistically significant conclusion especially when CNNs/deep learning is used. For statically significant conclusions, the proposed methodology must be tested with signficantly large datsets. This will rule-out over-fitting as well.

8. The results comparison with [80-84] is completely unfair. All of these are published between 2011-2017 and there is no justification of NOT including anything which was published in last five years.

9. Similarly, comparison with the methodologies presented in [7,9,12,15] is also unfair as most of them are conference works. When targeting a high impact journal like Sensors, the comparison is expected to be with high quality works. Authors must include latest high impact works for comparison.

Round 2

Reviewer 3 Report

Many of the concerns I raised in previous revision are addressed in the updated manuscript. The article may be accepted for publication.